# ENTROPYLONG: EFFECTIVE LONG-CONTEXT TRAINING VIA PREDICTIVE UNCERTAINTY

**Junlong Jia**[1,5]**, Ziyang Chen**[2]**, Xing Wu**[2✉]**, Chaochen Gao**[2]**, Zijia Lin**[3]**, Songlin Hu**[2]
**, Binghui Guo**[1,4,5✉]

[1]School of Artificial Intelligence, Beihang University
[2]Institute of Information Engineering, Chinese Academy of Sciences
[3]Tsinghua University
[4]Beijing Advanced Innovation Center for Future Blockchain and Privacy Computing, Beijing
[5]LMIB, NLSDE, Beihang University, Beijing
`{jiajunlong,guobinghui}@buaa.edu.cn`
`{gaochaochen,wuxing,husonglin,chenziyang}@iie.ac.cn`
`linzijia07@tsinghua.org.cn`

## ABSTRACT

Training long-context language models to capture long-range dependencies requires specialized data construction. Current approaches, such as generic text concatenation or heuristic-based variants, frequently fail to guarantee genuine long-range dependencies. We propose **EntropyLong**, a novel data construction method that leverages predictive uncertainty to verify dependency quality. Our approach identifies high-entropy positions in documents, retrieves semantically relevant contexts from large corpora, and verifies their utility by assessing whether they reduce prediction entropy. This *model-in-the-loop verification* ensures each dependency represents measurable information gain rather than spurious correlation. We construct training samples with long-range dependencies by combining original documents with these verified contextual supplements. Using FineWeb-Edu and Cosmopedia, we generate a dataset of 128K-length sequences with verified dependencies. Models trained on this data demonstrate significant improvements on RULER benchmarks, particularly in tasks requiring distant information. Following instruction fine-tuning, our models also achieve substantial gains on LongBench-v2, demonstrating enhanced long-context understanding. Extensive ablation studies further validate the necessity and effectiveness of entropy-based verification for long-context training.

## 1 INTRODUCTION

Processing and reasoning over extensive information spans represents a fundamental challenge for Large Language Models (LLMs) (Huo et al., 2025; Liu et al., 2024a; Yang et al., 2025). While architectural innovations such as Longformer (Beltagy et al., 2020), Big Bird (Zaheer et al., 2020), and rotary position embeddings (Su et al., 2024; Peng et al., 2023; Ding et al., 2024) have expanded theoretical context windows to millions of tokens, a fundamental bottleneck persists: the scarcity of training data that genuinely enables effective utilization of this extended context (Liu et al., 2023; An et al., 2024). The predominant practice (Chen et al., 2023) of concatenating short, unrelated documents frequently fails to establish meaningful long-range dependencies, resulting in models with extended attention spans but limited capability to leverage them effectively.

To address this data scarcity, recent research (Gao et al., 2024; 2025) has focused on synthesizing long-context training examples through heuristic principles. Query-centric methods like Quest (Gao et al., 2024) construct sequences by retrieving semantically related documents for topical coherence. Discrimination-driven approaches like NExtLong (Gao et al., 2025) interleave relevant documents with distractors to enhance long-distance information discrimination. Task-driven synthesis (Wei et al., 2024; Nadăş et al., 2025) generates data for specific long-context skills, while self-generated synthesis (Li et al., 2024) leverages LLMs to create their own training data. Despite significant

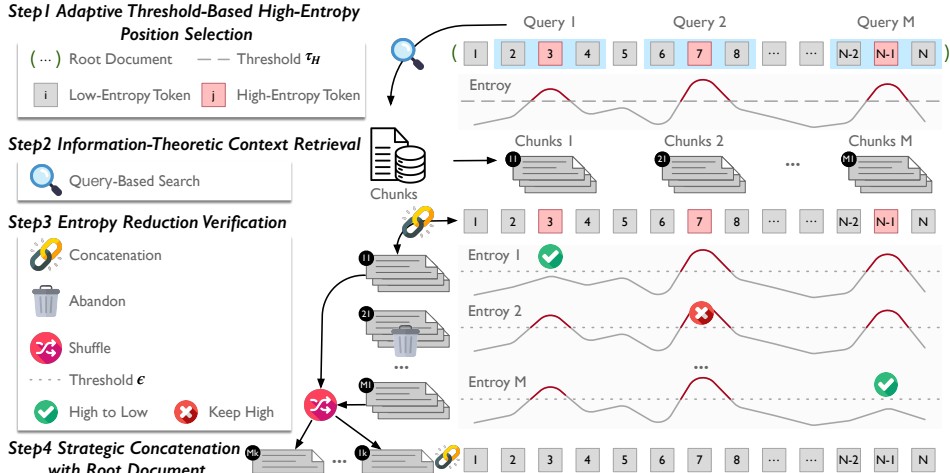

Figure 1: Overview of the EntropyLong framework. **Step 1. Adaptive Threshold-Based High-Entropy Position Selection**: Identify high-entropy tokens (red) exceeding the adaptive threshold and generate queries for uncertain positions. **Step 2. Information-Theoretic Context Retrieval**: Retrieve relevant document chunks from large corpora using query-based search. **Step 3. Entropy Reduction Verification**: Verify whether retrieved chunks reduce entropy - retain successful chunks (green checkmark) and discard ineffective ones. **Step 4. Strategic Concatenation**: Shuffle verified chunks and concatenate with the root document to create training sequences with validated dependencies.

progress, these strategies share a critical limitation: they presuppose what constitutes a "good" long-context example *without directly verifying its utility from the model's perspective*.

This paper introduces EntropyLong, a new paradigm for long-context data construction that replaces heuristic-based synthesis with principled, *model-in-the-loop verification*. Our key insight is that *a model's predictive uncertainty (entropy) directly signals an information deficit*. High-entropy positions—where the model struggles to predict the next token—mark locations where additional context may resolve uncertainty. When introducing relevant information from far earlier in the sequence reduces such entropy, this creates a genuine long-range dependency: accurate prediction requires integrating distant context. This establishes a principled connection between information deficits, verified entropy reduction, and effective long-range dependencies, drawing inspiration from active learning (Settles, 2009) and uncertainty-guided data curation (Pang et al., 2024). EntropyLong operationalizes this insight through a four-stage pipeline: (1) identifying high-entropy positions using adaptive thresholding, (2) retrieving semantically relevant contexts for these positions, (3) empirically verifying that retrieved contexts reduce prediction entropy, and (4) concatenating verified contexts with original documents to create training sequences with long-range dependencies.

We validate our approach by applying this methodology to FineWeb-Edu (Lozhkov et al.) and Cosmopedia (Allal et al.), constructing a comprehensive 128K-length training corpus with verified long-range dependencies. Our experimental evaluation demonstrates that models trained with EntropyLong substantially outperform baseline approaches on RULER (Hsieh et al., 2024) benchmarks across 8K-128K context lengths. Following instruction tuning, our approach achieves substantial performance gains on LongBench-v2 (Bai et al., 2024). We further conduct comprehensive ablation studies to validate the necessity and effectiveness of entropy-based verification.

Our primary contributions are:

1. **EntropyLong Framework**: We propose a novel data construction paradigm using model-in-the-loop verification to identify high-entropy positions and empirically verify that distant context reduces prediction uncertainty, establishing genuine long-range dependencies through information-theoretic validation.

2. **High-Quality Long-Context Dataset**: We construct and plan to open-source a 128K-length training corpus from FineWeb-Edu and Cosmopedia, containing samples with verified long-range dependencies that demonstrate measurable entropy reduction.

3. **Comprehensive Validation**: We achieve substantial performance improvements on RULER and LongBench-v2, with ablation studies confirming the necessity and effectiveness of entropy-based verification for long-context training.

## 2 RELATED WORK

Our work integrates two primary research areas: creating data for long-context model training and using predictive uncertainty for data curation.

### 2.1 DATA CONSTRUCTION FOR LONG-CONTEXT TRAINING

Early long-context training methods (Xiong et al., 2023; Glm et al., 2024) often use curriculum learning, training models on progressively longer sequences, for instance by simple document concatenation. However, this may not create genuine long-range dependencies, leading to more advanced synthetic data generation techniques. **Task-Driven Synthesis:** This method (Gao et al., 2025) generates data for specific long-context skills. A prominent example adapts the "Needle-in-a-Haystack" (NIAH) (Kamradt, 2023) evaluation to create training data that teaches models to find facts in distractor text (Liu et al., 2023). Other work synthesizes long-form question-answering datasets to enhance multi-step reasoning (Nadăş et al., 2025). **Coherence-Driven Synthesis:** This approach (Jiang et al., 2023; Shi et al., 2024; Gao et al., 2024; 2025) focuses on creating semantically fluent long documents by merging topically related texts. Approaches like Quest exemplify this by generating interconnected texts that demand long-range understanding, assuming topical relevance fosters useful dependencies (Gao et al., 2024). **Self-Generated Synthesis:** An emerging method involves an LLM generating its own long-context training data (Li et al., 2024). This offers a scalable solution that bypasses data scarcity and licensing issues associated with external teacher models. **Verification-Driven Synthesis:** A recent method is RE³SYN (Zhang et al., 2025), which retrieves similar documents and infers dependencies by reordering candidates using perplexity from a small proxy model. This doc-level, proxy criterion is misaligned with the target LM's *parametric knowledge* [1] and coarse in granularity. In contrast, EntropyLong keeps the target model in the loop: it locates token-level uncertainty via predictive entropy and retains only contexts that empirically reduce it. This position-level, model-aligned test guarantees measurable information gain, suppresses pseudo-dependencies and redundancy, and scales without permutation search.

### 2.2 PREDICTIVE UNCERTAINTY AS A GUIDING SIGNAL

A model's predictive uncertainty is a foundational signal in machine learning, most notably used in active learning to select the most informative data for human annotation, thereby improving efficiency (Settles, 2009). The principle is that models learn most from data they are uncertain about. In the LLM era, model feedback is vital for data curation, especially in teacher-student setups for alignment and instruction tuning (Pang et al., 2024). Uncertainty also guides active preference learning to request human feedback efficiently. While entropy improves reliability in retrieval-augmented generation, its application to the *unsupervised construction of pre-training data* is less explored. Some research uses predictive difficulty diagnostically, for instance, to design better evaluation metrics like LongPPL (Fang et al., 2024). In contrast, EntropyLong uses this signal constructively. It creates a closed loop where the model's learning state directs the generation of data it needs, grounding curation in information-theoretic principles. This self-supervised approach differs from teacher-student models by allowing a model to autonomously refine its pre-training data, paving the way for more automated data curation pipelines.

## 3 THEORETICAL MOTIVATION FOR ENTROPY-DRIVEN DATA CONSTRUCTION

Before detailing our methodology, we establish the theoretical framework that motivates EntropyLong. We argue that a model's own predictive uncertainty is the most direct and reliable signal for identifying information deficits. By leveraging this signal, we can move beyond heuristic-based

---

[1] As transfer to the target LM.

data construction towards a more principled, information-theoretic approach. Our framework is built upon a core premise, from which we derive a set of guiding principles and testable hypotheses.

## 3.1 THE PREMISE: PREDICTIVE ENTROPY AS A SIGNAL FOR INFORMATION DEFICIT

Our central premise is that high predictive uncertainty at a specific position may reflect an **information deficit**. The model lacks sufficient context to make a confident prediction. This uncertainty can be precisely quantified using Shannon Entropy (Shannon, 1948). Given a language model $M$ with parameters $\boldsymbol{\theta}$, the entropy at position $t$ is:

$$H_{\boldsymbol{\theta}}(x_t|x_{<t}) = -\sum_{v \in \mathcal{V}} P_{\boldsymbol{\theta}}(v|x_{<t}) \log P_{\boldsymbol{\theta}}(v|x_{<t}) \tag{1}$$

where $\mathcal{V}$ denotes the vocabulary set of all possible tokens. A position $t$ with high entropy, which we term a **High-Uncertainty Position**, thus serves as a natural anchor point—a location where external context could be maximally beneficial.

This premise shifts the focus from curating data based on human-defined heuristics (e.g., semantic similarity) to directly addressing the model's expressed needs. The challenge, then, is to find and verify the context that most effectively resolves this uncertainty.

## 3.2 THE PRINCIPLE: MAXIMIZING INFORMATION GAIN THROUGH EMPIRICAL VERIFICATION

From our core premise, we derive a guiding principle: *effective long-context data should be constructed by introducing distant information that demonstrably reduces the model's predictive uncertainty at points of information deficit.* We formalize this principle using the concept of Contextual Information Gain.

Given a document $D$ with a high-entropy position $t$ and a candidate context $C$, the **Contextual Information Gain** $\Delta I_t(C, D)$ measures the relative reduction in entropy after prepending $C$:

$$\Delta I_t(C, D) = \frac{H_{\boldsymbol{\theta}}(x_t|x_{<t}^D) - H_{\boldsymbol{\theta}}(x_t|x_{<t+|C|}^{[C;D]})}{H_{\boldsymbol{\theta}}(x_t|x_{<t}^D)} \tag{2}$$

where $x_{<t+|C|}^{[C;D]}$ denotes the context from the concatenated sequence $[C; D]$.

This principle has two practical implications for data construction:

1. **Empirical Verification**: We must empirically confirm that a candidate context provides a meaningful information gain (i.e., $\Delta I_t > \epsilon$) before accepting it as a valid long-range dependency. This verification step is the cornerstone of our approach, ensuring that every constructed dependency is genuinely useful to the model.

2. **Information-Theoretic Selection**: The optimal context $C^*$ is not merely one that is topically related, but one that maximizes the information gain: $C^* = \arg\max_{C \in \mathcal{R}} \Delta I_t(C, D)$. This moves beyond similarity metrics to a more direct measure of utility.

## 3.3 TESTABLE HYPOTHESES

This principled framework leads to a set of clear, empirically testable hypotheses that guide our experimental validation:

- **Hypothesis 1 (Necessity of Verification)**: We hypothesize that the empirical verification of entropy reduction is crucial. Training data built with verified contexts will significantly outperform data constructed using semantic retrieval alone, especially on tasks requiring fine-grained, long-range dependencies.

- **Hypothesis 2 (Optimality of Thresholds)**: We hypothesize that the thresholds for identifying high-uncertainty positions and for verifying information gain are critical. Optimal thresholds exist that balance the trade-off between the quality of dependencies and the quantity of available training signals.

These hypotheses are systematically tested in our analysis (Section 6), where we validate the importance of verification (Section 6.1) and the impact of thresholding (Section 6.2).

# 4 METHODOLOGY

We present the detailed methodology of EntropyLong. Our approach operationalizes the theoretical framework from Section 3 through four main stages: (1) adaptive threshold-based high-entropy position selection, (2) information-theoretic context retrieval, (3) entropy reduction verification, and (4) strategic concatenation with root documents. The complete algorithm is provided in Appendix B.

## 4.1 ADAPTIVE THRESHOLD-BASED HIGH-ENTROPY POSITION SELECTION

We use document-specific entropy statistics to identify high-uncertainty positions. Given a document $D = \{x_1, x_2, \ldots, x_n\}$, we pass it through a base language model $M_{\boldsymbol{\theta}}$ to compute the predictive entropy at each position using Equation 1.

We identify high-entropy positions using an adaptive threshold based on the document's entropy distribution:

$$\tau_H = \mu_H + \alpha \sigma_H \tag{3}$$

where $\mu_H$ and $\sigma_H$ are the mean and standard deviation of entropy values within the document, and $\alpha$ is a selectivity parameter (we use $\alpha = 2.0$ in practice).

Positions where $H_{\boldsymbol{\theta}}(x_t|x_{<t}) > \tau_H$ are marked as *high-entropy positions* $\mathcal{U} = \{t_1, t_2, \ldots, t_k\}$.

## 4.2 INFORMATION-THEORETIC CONTEXT RETRIEVAL

For each identified high-entropy position $t_i$, we use the surrounding context to retrieve relevant content from the corpus. The surrounding context is composed of $w$ words before and after the high-entropy position:

$$q_i = x_{t_i - w : t_i + w} \tag{4}$$

where $w$ is the context window size (we use $w = 16$ in practice).

Using this surrounding context as a query, we retrieve potentially relevant documents from a large pretraining corpus $\mathcal{R}$ using dense retrieval. We employ a pre-trained sentence transformer (Sturua et al., 2024) to encode both the query and candidate documents, then retrieve the top-$K$ most similar documents based on cosine similarity:

$$\text{sim}(q_i, d_j) = \frac{\text{embed}(q_i) \cdot \text{embed}(d_j)}{|\text{embed}(q_i)| \cdot |\text{embed}(d_j)|} \tag{5}$$

## 4.3 ENTROPY REDUCTION VERIFICATION FOR RETRIEVED CONTENT

We empirically verify that retrieved contexts actually reduce model uncertainty. For each retrieved candidate context $C_j$, we prepend it to the original document and re-evaluate the model's entropy at the high-entropy position:

$$H'_{\boldsymbol{\theta}}(x_{t_i}|x^{[C_j;D]}_{<t_i+|C_j|}) = -\sum_{v \in \mathcal{V}} P_{\boldsymbol{\theta}}(v|x^{[C_j;D]}_{<t_i+|C_j|}) \log_2 P_{\boldsymbol{\theta}}(v|x^{[C_j;D]}_{<t_i+|C_j|}) \tag{6}$$

We only retain context $C_j$ if it satisfies the entropy reduction threshold:

$$\Delta I_{t_i}(C_j, D) = \frac{H_{\boldsymbol{\theta}}(x_{t_i}|x^D_{<t_i}) - H'_{\boldsymbol{\theta}}(x_{t_i}|x^{[C_j;D]}_{<t_i+|C_j|})}{H_{\boldsymbol{\theta}}(x_{t_i}|x^D_{<t_i})} > \epsilon \tag{7}$$

where we use $\epsilon = 0.4$.

## 4.4 STRATEGIC CONCATENATION WITH ROOT DOCUMENTS

For each original document $D$ with verified contexts $\{C_1, C_2, \ldots, C_m\}$, we construct training sequences through strategic concatenation. We employ two main strategies:

**Shuffle Strategy:** We randomly shuffle the retrieved contexts (which correspond to different high-entropy positions in the original token order) before concatenating them with the root document.

Table 1: Main results on the RULER benchmark. Best results are in **bold**.

| RULER | 8k | 16k | 32k | 64k | 128k | avg |
|---|---|---|---|---|---|---|
| Quest | 91.39 | 89.72 | 84.37 | 77.07 | 60.11 | 80.53 |
| NExtLong | 89.99 | 88.58 | 86.04 | 83.52 | 77.99 | 85.22 |
| **EntropyLong** | **91.50** | **90.11** | **88.95** | **85.04** | **81.26** | **87.37** |

**Sequence Strategy:** We organize the retrieved contexts according to the original token order of their corresponding high-entropy positions in the document.

The final training sample is constructed as:

$$S = [C_{\pi(1)}; C_{\pi(2)}; \ldots; C_{\pi(m)}; D] \tag{8}$$

where $\pi$ is a random permutation of $\{1, 2, \ldots, m\}$ for the shuffle strategy, or the identity permutation for the sequence strategy. We use the random shuffle strategy as it demonstrates better performance in our experiments.

## 5 EXPERIMENTS

We conduct a comprehensive set of experiments to evaluate the effectiveness of EntropyLong. Our evaluation aims to answer two key questions: (1) Does EntropyLong improve performance on long-context benchmarks compared to existing data construction strategies? (2) Do these improvements during pretraining translate to better performance on downstream tasks after instruction fine-tuning?

### 5.1 EXPERIMENTAL SETUP

**Base Model and Training.** We use Meta-Llama-3-8B (Dubey et al., 2024) as our base model, extending the context window to 128K tokens. Following NExtLong (Gao et al., 2025), we only modify the RoPE (Rotary Position Embedding) base value to 200,000,000 to achieve this context extension. The model is trained for 1000 iterations with a global batch size of 4M tokens. Complete training hyperparameters are provided in Appendix C.

**Dataset Construction.** We construct the EntropyLong dataset using FineWeb-Edu (Lozhkov et al.) and Cosmopedia (Allal et al.). We sample 100K documents as source texts and use the full corpora (totaling over 1B documents) for retrieval. Our methodology generates 4B tokens of 128K-length sequences, where each sequence contains verified long-range dependencies with an average information gain of $\bar{\Delta I} = 0.68$ per dependency. Implementation details are provided in Appendix D.

**Baselines.** We compare against two leading methods as baselines, using identical Llama-3-8B training configurations and 4B tokens. *Quest (Gao et al., 2024):* Coherence-driven retrieval and concatenation of semantically similar documents. *NExtLong (Gao et al., 2025):* Discrimination-driven document interleaving with distracting negative examples.

**Evaluation Benchmarks.** We evaluate on two main benchmarks: (1) RULER (Hsieh et al., 2024), which includes needle-in-a-haystack, multi-hop reasoning, variable tracking, and pattern extraction tasks at 8K, 16K, 32K, 64K, and 128K context lengths; (2) LongBench-v2 (Bai et al., 2024) after instruction fine-tuning, which tests real-world long-context capabilities across diverse domains. Complete benchmark descriptions are in Appendix F.

### 5.2 MAIN RESULTS

### 5.2.1 RULER BENCHMARK RESULTS

As shown in Table 1, EntropyLong achieves superior performance on the RULER benchmark across different context lengths. Our model consistently outperforms baselines, with particularly strong results at longer context. EntropyLong achieves an average score of 87.37 across all context lengths, obviously outperforming Quest (80.53) and NExtLong (85.22). On 128k context length, EntropyLong achieves a score of 81.26, significantly outperforming Quest (60.11) and NExtLong (77.99).

Table 2: Results on LongBench-v2 after instruction fine-tuning (UltraChat (Ding et al., 2023)).

| Model | Easy | Hard | Short | Medium | Long | Overall |
|---|---|---|---|---|---|---|
| Quest | 17.70 | 25.10 | 25.60 | 20.00 | 21.30 | 22.30 |
| NExtLong | 21.40 | 25.70 | 27.20 | 21.90 | 23.10 | 24.10 |
| **EntropyLong** | **25.50** | **28.90** | **30.00** | **23.70** | **31.50** | **27.60** |

Table 3: The verification step is critical for performance. We report RULER scores across different context lengths and the average.

| RULER | 8k | 16k | 32k | 64k | 128k | avg |
|---|---|---|---|---|---|---|
| EntropyLong-NoVerify | 91.44 | 88.76 | 86.29 | 83.85 | 79.47 | 85.82 |
| **EntropyLong** *(Full Method)* | **91.50** | **90.11** | **88.95** | **85.04** | **81.26** | **87.37** |

### 5.2.2 INSTRUCTION FINE-TUNING RESULTS

The benefits of our pretraining strategy carry over strongly to downstream tasks after supervised fine-tuning (SFT). As shown in Table 2, our model achieves significant improvements on LongBench-v2 across different task categories. EntropyLong demonstrates particularly strong performance on Long tasks (31.50) compared to Quest (21.30) and NExtLong (23.10), showing a significant improvement of 8.40 points over the best baseline. This indicates that the model has learned to effectively locate and utilize information from extensive contexts. These results confirm that building a pretraining dataset with empirically verified dependencies leads to more capable long-context models.

The superior performance can be attributed to the quality of our constructed dependencies. Entropy-Long dataset achieves an average information gain of $\Delta I = 0.68$ per verified dependency, indicating substantial uncertainty reduction compared to unverified retrieval. Examining the performance gains more closely, we observe that EntropyLong's advantage over NExtLong scales progressively with context length on RULER. This non-uniform scaling pattern provides empirical evidence that our information-theoretic approach to data construction translates to performance improvements.

## 6 ANALYSIS

This section provides a comprehensive empirical validation of our theoretical framework from Section 3. We examine each of our two core hypotheses through targeted experiments: (1) the necessity of empirical verification for effective data construction (Hypothesis 1), and (2) the existence of optimal threshold parameters (Hypothesis 2). Additionally, we conduct detailed ablation studies on key hyperparameters and provide qualitative analysis of the learned attention patterns.

### 6.1 VERIFICATION NECESSITY (HYPOTHESIS 1)

We test Hypothesis 1 by comparing our full method against "EntropyLong-NoVerify", which skips empirical verification and always accepts the top-retrieved document based on semantic similarity alone. Table 3 strongly supports Hypothesis 1. The full EntropyLong method achieves 87.37 vs. 85.82 for the non-verified version (+1.55 points), with consistent improvements across all context lengths: 16k (+1.35), 32k (+2.66), 64k (+1.19), and 128k (+1.79). The performance gaps demonstrate that empirical verification is essential for effective long-context training data, as it filters out spurious correlations that may degrade model performance.

### 6.2 THRESHOLD OPTIMALITY (HYPOTHESIS 2)

We validate Hypothesis 2 by examining how two key threshold parameters affect the balance between data quality and data quantity: the high-entropy position selection threshold ($\tau_H$) and the entropy reduction verification threshold ($\epsilon$). Our hypothesis predicts that optimal threshold values exist to achieve the best balance between these competing factors.

**High-Entropy Position Selection Threshold ($\tau_H$)** The parameter $\alpha$ controls how many high-entropy positions we select: lower values select more positions (more data) while higher values select fewer but more selective positions (higher quality). Table 4 confirms our hypothesis. When

Table 4: Performance on RULER benchmark with varying adaptive threshold $\alpha$ values. #Tokens indicates the number of high-entropy tokens selected for context retrieval. Higher $\alpha$ values result in fewer but more selective high-entropy positions.

| RULER | #Tokens | 8k | 16k | 32k | 64k | 128k | avg |
|---|---|---|---|---|---|---|---|
| EntropyLong ($\alpha = 1.5$) | 913 | 90.25 | 87.16 | 82.07 | 79.49 | 73.47 | 82.49 |
| **EntropyLong ($\alpha = 2.0$)** | 292 | **91.50** | **90.11** | **88.95** | **85.04** | **81.26** | **87.37** |
| EntropyLong ($\alpha = 2.5$) | 83 | 90.96 | 88.29 | 85.52 | 82.83 | 80.02 | 85.52 |

Table 5: Performance on RULER benchmark with varying entropy reduction thresholds. #Verified indicates the number of verified dependencies retained after entropy reduction validation. The threshold of 0.4 provides the best empirical performance.

| RULER | #Verified | 8k | 16k | 32k | 64k | 128k | avg |
|---|---|---|---|---|---|---|---|
| EntropyLong ($\epsilon = 0.2$) | 62 | 91.44 | 88.44 | 84.97 | 83.53 | 78.84 | 85.45 |
| **EntropyLong ($\epsilon = 0.4$)** | 46 | **91.50** | 90.11 | **88.95** | **85.04** | 81.26 | **87.37** |
| EntropyLong ($\epsilon = 0.6$) | 29 | 90.31 | 88.64 | 86.19 | 84.09 | **81.46** | 86.14 |
| EntropyLong ($\epsilon = 0.8$) | 13 | 91.39 | **90.29** | 87.60 | 83.90 | 79.19 | 86.47 |

$\alpha = 1.5$, we get many positions (913 tokens) but performance drops (82.49 vs 87.37) due to noisy selections. When $\alpha = 2.5$, we get very few positions (83 tokens) with high quality but insufficient training data. The optimal $\alpha = 2.0$ (292 tokens) strikes the right balance between having enough high-quality training signals.

**Entropy Reduction Verification Threshold ($\epsilon$)**   The threshold $\epsilon$ determines how strict we are when verifying that retrieved contexts actually help reduce uncertainty. Table 5 strongly supports Hypothesis 2. With a low threshold ($\epsilon = 0.2$), we keep many contexts (62 verified dependencies) but accept weak ones that barely help, hurting performance (85.45 vs 87.37). With high thresholds ($\epsilon = 0.6, 0.8$), we only keep very helpful contexts but get too few for effective training (29, 13 dependencies). The optimal $\epsilon = 0.4$ (46 dependencies) gives us the best balance: enough high-quality verified contexts for effective training.

Finding clear optimal values for both $\alpha$ (2.0) and $\epsilon$ (0.4) strongly validates Hypothesis 2. These results show that our method works best when we carefully tune these thresholds to balance data quality and quantity. This confirms that our information-theoretic approach is both theoretically sound and practically effective.

## 6.3   IMPACT OF DIFFERENT WINDOW SIZES $w$

The window size $w$ determines how much surrounding context is used to retrieve relevant documents from the corpus. We tested different values of $w$ to understand its impact on model performance. The window size analysis reveals an interesting non-monotonic relationship between context richness and retrieval effectiveness. While larger windows ($w = 16$) generally provide richer contextual information leading to better retrieval quality, the relationship is not strictly monotonic—$w = 8$ shows superior performance at shorter contexts (8k, 16k) but degrades at longer contexts. This suggests that the optimal window size may depend on the target context length. Our setting of $w = 16$ achieves the best overall balance across the full range of context lengths.

## 6.4   IMPACT OF DIFFERENT CONCATENATION STRATEGIES

We compare two different strategies for concatenating retrieved contexts with root documents: sequence-based and random shuffle-based concatenation. The random shuffle strategy slightly outperforms the sequence-based approach. Random shuffling helps prevent the model from learning a positional bias tied to the relative ordering of supplementary documents. As a result, the model is forced to search the entire context to locate relevant evidence rather than relying on shortcut positional cues. This reliance on global retrieval becomes especially beneficial at long context lengths (e.g., 32K and 128K), which likely explains why the random shuffle variant yields noticeably larger gains in these settings.

Table 6: Performance on RULER benchmark with varying window sizes $w$. Larger windows provide more context for retrieval but may introduce noise.

|  | 8k | 16k | 32k | 64k | 128k | avg |
|---|---|---|---|---|---|---|
| EntropyLong ($w = 2$) | 91.12 | 89.55 | 88.16 | 84.49 | 79.75 | 86.61 |
| EntropyLong ($w = 4$) | 90.91 | 88.91 | 86.96 | 84.39 | 79.03 | 86.04 |
| EntropyLong ($w = 8$) | **92.34** | **91.01** | 86.97 | 83.86 | 80.06 | 86.85 |
| **EntropyLong** ($w = 16$) *(current setting)* | 91.50 | 90.11 | **88.95** | **85.04** | **81.26** | **87.37** |

Table 7: Performance comparison between different concatenation strategies. Random strategy shows better performance than sequence-based approach.

| RULER | 8k | 16k | 32k | 64k | 128k | avg |
|---|---|---|---|---|---|---|
| EntropyLong (Sequence) | **91.55** | **90.91** | 87.47 | 84.91 | 80.43 | 87.06 |
| **EntropyLong (Random)** *(current setting)* | 91.50 | 90.11 | **88.95** | **85.04** | **81.26** | **87.37** |

Table 8: Performance on short text benchmarks. EntropyLong maintains competitive performance on short texts while excelling at long contexts.

|  | arc_c | arc_e | hellaswag | piqa | logiqa | winogrande | avg |
|---|---|---|---|---|---|---|---|
| Llama3-8b-base | 50.34 | 80.18 | **60.13** | 79.60 | **27.50** | **72.85** | 61.77 |
| **+EntropyLong** | **51.45** | **80.98** | 59.61 | **80.47** | 26.27 | 72.22 | **61.83** |

## 6.5 SHORT TEXT PERFORMANCE

To demonstrate that our method does not negatively impact short text performance, we evaluate on several short text benchmarks. EntropyLong maintains very similar performance on short text tasks (61.83 vs. 61.77 average), demonstrating that our long-context training approach does not compromise overall short text capabilities.

## 6.6 ENTROPY REDUCTION FROM SUPPLEMENTARY DOCUMENTS

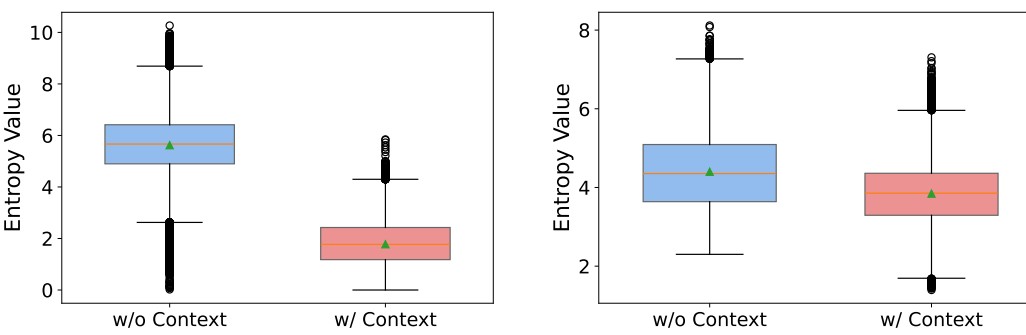

(a)                                     (b)

Figure 2: Distribution of entropy before and after adding the supplementary document that maximally reduces entropy for each high-entropy token. (a) shows the entropy distribution for tokens that pass verification before and after adding the retrieved document, (b) shows the corresponding entropy distribution for high-entropy tokens that fail verification before and after adding the retrieved document.

We analyze the entropy distribution of high-entropy tokens before and after adding, for each token, the retrieved document that yields the largest entropy reduction, as shown in Figure 2a and Figure 2b.

For verified high-entropy tokens, the average entropy before adding any supplementary document is 5.62. After incorporating the document that maximally reduces entropy for each token, the average entropy drops to 1.70, indicating that verified supplementary documents substantially resolve local uncertainty at these positions.

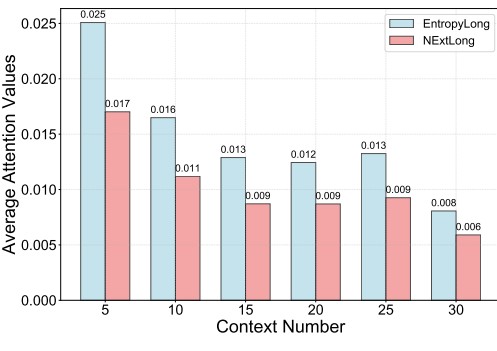 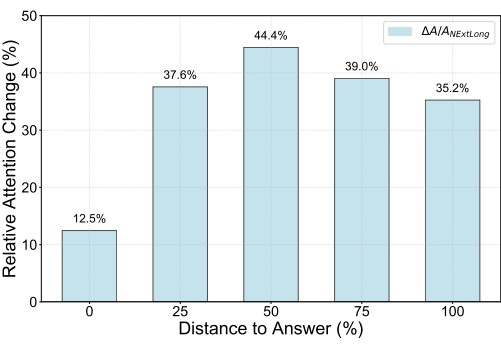

(a)                 (b)

Figure 3: EntropyLong's attention patterns analysis. (a) Attention to correct answers vs NExtLong across different context chunks with answers at front; (b) Relative attention vs NExtLong with answers at different positions, where $\Delta A$ represents the attention difference.

In contrast, for high-entropy tokens that fail verification, the average entropy before adding any supplementary document is 4.40. After incorporating the document that yields the largest entropy reduction among the retrieved candidates, the average entropy decreases only slightly to 3.84, indicating that supplementary documents provide limited information gain for these positions and do not meaningfully resolve the model's uncertainty.

### 6.7 ANALYSIS OF ATTENTION PATTERNS

To validate the effectiveness of our uncertainty-driven training approach, we further construct experimental sequences by concatenating answer, question, and irrelevant text chunks, then analyze attention patterns to correct answers for evaluating long-range dependency capabilities. Figure 3 reveals the differences in how EntropyLong and NExtLong process long-range dependencies.

**Context Length Scaling (Figure 3a):** We analyze attention allocation patterns by fixing correct answers at the sequence front and comparing attention scores between EntropyLong and NExtLong as context length increases. EntropyLong consistently maintains higher attention to correct answers than NExtLong, with the attention advantage remaining stable across increasing context lengths, demonstrating superior capability to focus on relevant information regardless of sequence length.

**Lost-in-the-Middle Mitigation (Figure 3b):** We evaluate positional bias by comparing attention scores to correct answers positioned at various locations within sequences. EntropyLong alleviates the "lost-in-the-middle" phenomenon, achieving substantial relative improvements over NExtLong: 12.5% (0%), 37.6% (25%), 44.4% (50%), 39.0% (75%), and 35.2% (100% distance). The peak improvements at middle positions (25%-75%) directly validate mitigation of this pervasive limitation in standard long-context models.

## 7 CONCLUSION

We introduce EntropyLong, a novel data synthesis method that leverages predictive uncertainty to construct high-quality long-context training data. Our key insight is that a model's uncertainty signals where additional context is most beneficial, enabling construction of training datasets with verified functional dependencies rather than heuristic similarity. Experiments show superior performance on RULER and LongBench-v2, with ablations confirming empirical verification is essential. EntropyLong shifts from heuristic-based to evidence-based data construction, offering a more reliable path toward robust long-context understanding.

## 8 ACKNOWLEDGMENTS

This work is supported by the Brain Science and Brain-like Intelligence Technology-National Science and Technology Major Project (Grant No. 2021ZD0201302).

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

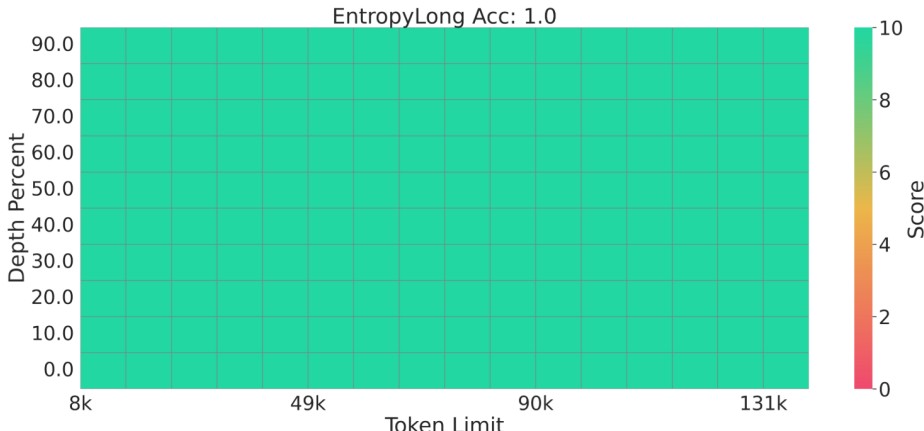

Figure 4: EntropyLong's performance on the needle-in-a-haystack task within a 128K context window. The heatmap shows accuracy across different text lengths and needle positions, with darker colors indicating higher accuracy. EntropyLong achieves perfect accuracy across all configurations, successfully locating the target information regardless of position within the context.

## A  NEEDLE-IN-A-HAYSTACK ANALYSIS

To further validate EntropyLong's effectiveness in long-context understanding, we conducted experimental validation on the classic "Needle-in-a-Haystack" task (Kamradt, 2023). This task requires the model to accurately locate and extract specific information within large amounts of irrelevant text, serving as an important benchmark for evaluating long-context model performance.

As shown in Figure 4, EntropyLong demonstrates exceptional performance on the needle-in-a-haystack task. The experimental results reveal:

**Perfect Accuracy Across All Positions:** EntropyLong achieves 100% accuracy in locating the target information regardless of where it appears within the 128K context window, from the very beginning to the end of the sequence.

**Length Robustness:** The model maintains perfect performance across different context lengths within the 128K window, demonstrating that our uncertainty-driven training approach effectively handles varying sequence lengths without degradation.

**Position Independence:** Unlike traditional long-context models that often suffer from the "lost-in-the-middle" problem, EntropyLong shows no performance drop when the target information is positioned in the middle of long sequences, indicating robust attention mechanisms across all positions.

These results confirm that EntropyLong's uncertainty-driven data construction method enables the model to learn effective long-range dependencies, resulting in superior performance on tasks requiring precise information retrieval from extended contexts.

## B  ALGORITHM DETAILS

This section provides the complete implementation details for reproducing our EntropyLong framework.

After selecting dependency positions under different thresholds, we obtain a set of verified supplementary context segments $\{C_i\}$ along with their corresponding root document $D$. To strictly maintain a total sample length of $L_{\text{target}} = 128K$ tokens, we allocate and fill a context budget as follows:

(1) **Compute the remaining budget for supplementary context.** For each root document, we compute the maximum budget available for its supplementary contexts:

$$B_{\text{supp}} = L_{\text{target}} - |D|, \tag{9}$$

where $|D|$ denotes the token length of the root document and $B_{\text{supp}}$ is the total length available for all supplementary context segments.

(2) **Sort supplementary segments by entropy reduction.** We rank all candidate segments in descending order according to their entropy-reduction value at the target high-entropy positions, so that the most informative segments are added first.

(3) **Fill the budget segment by segment.** We iterate through the sorted list and add segments one by one, updating the remaining budget after each addition.

If adding the next segment would exceed $B_{\text{supp}}$, we stop and discard the remaining segments, ensuring that the final sample does not exceed $L_{\text{target}} = 128K$ tokens. Finally, if the total length of "all available supplementary context + root document" is still below $L_{\text{target}}$ (i.e., not long enough to form a sufficiently long training instance), we discard the sample entirely and do not include it in the training set.

Algorithm 1 provides the complete pseudocode for our EntropyLong data construction framework. The algorithm implements our four-stage pipeline: identifying high-entropy positions through adaptive thresholding, retrieving semantically relevant contexts, empirically verifying entropy reduction, and constructing training sequences through strategic concatenation. The key innovation lies in the verification step, where we ensure that each retrieved context demonstrably reduces prediction uncertainty before inclusion in the final training data. This principled approach distinguishes EntropyLong from heuristic-based methods by grounding data construction decisions in measurable information-theoretic criteria.

---

**Algorithm 1** EntropyLong Data Construction

---

**Require:** Training corpus $\mathcal{D} = \{D_1, D_2, \ldots, D_N\}$, retrieval corpus $\mathcal{R}$, base model $M_{\boldsymbol{\theta}}$, target length $L_{\text{target}}$
**Ensure:** Synthesized long documents $\mathcal{S}$
1: Initialize $\mathcal{S} \leftarrow \emptyset$
2: **for** each document $D \in \mathcal{D}$ **do**
3:      $\mathcal{H} \leftarrow \text{ComputeEntropy}(D, M_{\boldsymbol{\theta}})$ {Compute entropy for all positions}
4:      $\tau_H \leftarrow \mu_{\mathcal{H}} + \alpha\sigma_{\mathcal{H}}$ {Adaptive threshold}
5:      $\mathcal{U} \leftarrow \{t : H_t > \tau_H\}$ {Identify uncertainty points}
6:      $\mathcal{C}_{\text{verified}} \leftarrow \emptyset$
7:      **for** each uncertainty point $t \in \mathcal{U}$ **do**
8:          $q \leftarrow x_{t-w:t+w}$ {Extract query context}
9:          $\mathcal{C}_{\text{candidates}} \leftarrow \text{Retrieve}(q, \mathcal{R}, K)$ {Retrieve top-K candidates}
10:         **for** each candidate $C \in \mathcal{C}_{\text{candidates}}$ **do**
11:            **if** $C \notin \mathcal{C}_{\text{verified}}$ **then**
               {Check for duplicates}
12:               $\Delta I \leftarrow \text{VerifyInformationGain}(C, D, t, M_{\boldsymbol{\theta}})$
13:               **if** $\Delta I > \epsilon$ **then**
14:                  $\mathcal{C}_{\text{verified}} \leftarrow \mathcal{C}_{\text{verified}} \cup \{C\}$
15:                  **break** {Use first verified context for this position}
16:               **end if**
17:            **end if**
18:         **end for**
19:      **end for**
20:      **if** $|\mathcal{C}_{\text{verified}}| > 0$ **then**
21:          $S \leftarrow \text{ConstructSequence}(\mathcal{C}_{\text{verified}}, D, L_{\text{target}})$
22:          $\mathcal{S} \leftarrow \mathcal{S} \cup \{S\}$
23:      **end if**
24: **end for**
25:
26: **return** $\mathcal{S}$

---

Table 9: Complete training hyperparameters for 128K model.

| Parameter | Value |
|---|---:|
| Initial Model | Meta-Llama-3-8B (Dubey et al., 2024) |
| rotary-emb-base | 200,000,000 |
| $\beta_1$ | 0.9 |
| $\beta_2$ | 0.95 |
| Learning Rate | $4 \times 10^{-5}$ |
| Precision | bfloat16 |
| Gradient Clipping | 1.0 |
| Weight Decay | 0.1 |
| LR Decay Style | cosine |
| Training Iterations | 1000 |
| Warmup Iterations | 200 |
| Sequence Length | 131,072 |
| Global Batch Size | 4M tokens |

## C  TRAINING CONFIGURATION

Table 9 presents the complete training hyperparameters used for our 128K context length models. We extend the Meta-Llama-3-8B base model by modifying only the RoPE base parameter to 200,000,000 to accommodate the extended context window. Our training uses a global batch size of 4M tokens with cosine learning rate decay and gradient clipping to ensure stable training on long sequences. These hyperparameters ensure fair comparison with baseline methods by following NExtLong's configuration.

## D  IMPLEMENTATION DETAILS

This section provides the complete implementation details for training with data generated by EntropyLong.

We detail all hyperparameters used in our entropy-driven data construction pipeline, including the adaptive thresholding mechanism, context retrieval system, and verification process. Additionally, we specify the configuration used for instruction fine-tuning to ensure complete reproducibility of our experimental results.

**EntropyLong Hyperparameters:**

- Query window size: $w = 16$ words
- Retrieval candidates: $K = 32$
- Entropy reduction threshold: $\epsilon = 0.4$
- Adaptive threshold parameter: $\alpha = 2.0$
- Sentence transformer: jina-embeddings-v3 (Sturua et al., 2024)
- Dense retrieval index: Faiss (Johnson et al., 2019) with 768-dimensional embeddings

**Instruction Tuning Configuration:**

- Dataset: UltraChat (Ding et al., 2023)
- Learning rate: $2 \times 10^{-5}$
- Batch size: 2M tokens
- Training Iterations: 250

## E  COMPUTATIONAL COST ANALYSIS

Table 10 summarizes the cost breakdown for both methods under the full 4B-token training setup.

We decompose the cost of long-context pretraining into two stages: (1) data construction and (2) model training, and compare EntropyLong against NExtLong, the strongest baseline in our experiments. For a fair comparison, we fully reproduce NExtLong's data construction pipeline. In the

Table 10: Computational cost breakdown for NExtLong and EntropyLong. "GPUh" denotes GPU hours; both models are trained on 4B tokens.

| Method | Embed | High-Ent. | Verify | Train | Total |
|---|---|---|---|---|---|
| NExtLong | 928 GPUh | – | – | 768 GPUh | 1696 GPUh |
| EntropyLong | 928 GPUh | 3 GPUh | 352 GPUh | 768 GPUh | 2051 GPUh |

Table 11: RULER performance when NExtLong and EntropyLong are matched under similar total GPU cost. NExtLong is trained on 6B tokens to equalize total GPU hours with EntropyLong (trained on 4B tokens).

| Method | 8k | 16k | 32k | 64k | 128k | avg | Total |
|---|---|---|---|---|---|---|---|
| NExtLong (6B tokens) | **91.97** | **90.52** | 85.81 | 81.88 | 78.60 | 85.75 | 2080 |
| EntropyLong (4B tokens) | 91.50 | 90.11 | **88.95** | **85.04** | **81.26** | **87.37** | 2051 |

data construction stage, EntropyLong incurs approximately 355 additional GPU hours relative to NExtLong, mainly due to per-candidate entropy-reduction verification.

However, to reach the same RULER performance as NExtLong, EntropyLong requires substantially fewer training tokens (2.4B vs. 4B), reducing the training cost by 307 GPU hours. This shows that EntropyLong is more token-efficient than NExtLong at matched performance.

We also examine a setting where both methods use a similar total computational budget. When increasing NExtLong's training tokens to 6B so that its total GPU cost reaches 2080 GPU hours, comparable to EntropyLong's 2051 GPU hours, its RULER performance reaches 85.75, which remains lower than EntropyLong's 87.37 under similar cost. As shown in Table 11, EntropyLong achieves higher performance at longer context lengths (32k to 128k). The results show that, when aiming for better performance—which typically needs more computational resources—EntropyLong demonstrates higher computational efficiency compared to previous methods.

# F   BENCHMARK DETAILS

**RULER Benchmark Tasks:**

- **Needle-in-a-Haystack (NIAH)**: Four retrieval-based tasks including Single NIAH (vanilla needle retrieval), Multi-keys NIAH (retrieving one needle among multiple distractors), Multi-values NIAH (retrieving all values with the same key), and Multi-queries NIAH (retrieving multiple distinct needles)
- **Variable Tracking (VT)**: Multi-hop tracing task that evaluates coreference resolution by tracking variable assignments through linear chains (e.g., X2 = X1, X3 = X2) and returning all variables pointing to the same initial value
- **Common Words Extraction (CWE) and Frequent Words Extraction (FWE)**: Aggregation tasks requiring models to identify the most frequent words from long sequences, serving as proxies for summarization capabilities where relevant information spans large portions of context
- **Question Answering (QA)**: Real-world adaptation of NIAH where golden paragraphs containing answers are inserted among distracting paragraphs from the same dataset, with questions serving as retrieval queries

**LongBench-v2 Tasks:**

- **Single-Document QA**: Question answering across six domains (Academic, Literary, Legal, Financial, Governmental, Detective) plus event ordering tasks, testing comprehension of individual long documents
- **Multi-Document QA**: Cross-document reasoning tasks requiring information synthesis from multiple sources across Academic, Legal, Financial, Governmental, and Multi-news domains

Table 12: Comparison between UTK and EntropyLong on the RULER benchmark.

| Method | 8k | 16k | 32k | 64k | 128k | avg |
|---|---|---|---|---|---|---|
| UTK | 89.51 | 84.66 | 80.23 | 77.84 | 71.98 | 80.84 |
| **EntropyLong** | **91.50** | **90.11** | **88.95** | **85.04** | **81.26** | **87.37** |

Table 13: RULER results for different verification thresholds $\sigma$ under a similar verified token numbers.

| Method | 8k | 16k | 32k | 64k | 128k | avg |
|---|---|---|---|---|---|---|
| $\sigma = 0.4$ | **90.73** | 88.08 | **85.63** | 82.16 | 77.93 | 84.91 |
| $\sigma = 0.6$ | 89.48 | **88.16** | 84.91 | **83.05** | **80.82** | **85.18** |

- **Long In-context Learning**: Three challenging tasks including user guide QA, new language translation (Zhuang and Kalamang), and many-shot learning with anonymized labels for classification tasks
- **Long-dialogue History Understanding**: Analysis of extended conversation histories including agent interaction games (GAMA-Bench) and multi-turn chat sessions (Long-MemEval) requiring memory of distant context
- **Code Repository Understanding**: Deep comprehension of large codebases requiring integration of multiple code components and understanding of complex software architectures
- **Long Structured Data Understanding**: Reasoning over extensive tabular data (financial reports) and large-scale knowledge graphs requiring multi-hop logical inference across interconnected entities

## G   ADDITIONAL RULER RESULTS WITH UTK BASELINE

For completeness, we also compare EntropyLong against the UTK (Tian et al., 2025) baseline on the RULER benchmark. Table 12 reports scores at different context lengths. EntropyLong consistently outperforms UTK across all lengths, with especially large gains at 16k–128k, highlighting the advantage of our entropy-verified dependency construction for very long contexts.

## H   EFFECT OF HIGHER VERIFICATION THRESHOLD $\sigma$ UNDER SIMILAR VERIFIED TOKEN NUMBERS

To verify our claim that the optimal context $C$ is the one that maximizes information gain while controlling for the effect of data quantity, we maintain an approximately constant number of verified tokens.

We then increase the verification threshold $\sigma$ to obtain a more strictly filtered subset, while keeping the number of verified dependency relations roughly similar to those identified under the original configuration. This results in a dataset of approximately 2.4B tokens, on which we train models with $\sigma = 0.4$ and $\sigma = 0.6$ and evaluate them on RULER (Table 13).

We observe that, under a comparable token budget, using a higher verification threshold ($\sigma = 0.6$) yields larger gains at longer context lengths (64k and 128k), confirming that more strictly verified dependencies can improve very-long-context performance.

## I   THE $\alpha-\epsilon$ TRADE-OFF AND ITS IMPACT ON MODEL PERFORMANCE

To further analyze the joint effect of $\alpha$ and $\epsilon$, we track RULER performance at different training steps for three configurations: a *high-$\alpha$, low-$\epsilon$* setting ($\alpha = 2.5$, $\epsilon = 0.2$), our final *balanced* setting ($\alpha = 2.0$, $\epsilon = 0.4$), and a *low-$\alpha$, high-$\epsilon$* setting ($\alpha = 1.5$, $\epsilon = 0.8$). The high-$\alpha$, low-$\epsilon$ setting yields a large set of weakly verified positions. This results in fast initial improvements but also leads to early saturation. In contrast, the other two settings produce similar improvement curves over time, with the balanced setting achieving consistently better final performance.

Table 14: Training dynamics of different $(\alpha, \epsilon)$ configurations on RULER at steps 200, 600, and 1000 (corresponding to 0.8B, 2.4B, and 4B tokens, respectively).

| Step / Setting | 8k | 16k | 32k | 64k | 128k | avg |
|---|---|---|---|---|---|---|
| *Step 200 (0.8B tokens)* | | | | | | |
| $\alpha = 2.5, \epsilon = 0.2$ | 90.58 | 88.95 | **85.43** | **80.94** | **78.74** | **84.93** |
| $\alpha = 2.0, \epsilon = 0.4$ | 90.90 | 87.03 | 83.13 | 77.82 | 72.99 | 82.37 |
| $\alpha = 1.5, \epsilon = 0.8$ | **91.33** | **89.57** | 82.75 | 79.22 | 70.13 | 82.60 |
| *Step 600 (2.4B tokens)* | | | | | | |
| $\alpha = 2.5, \epsilon = 0.2$ | **91.20** | **89.71** | **87.13** | **84.46** | 79.80 | **86.46** |
| $\alpha = 2.0, \epsilon = 0.4$ | 90.73 | 88.08 | 85.63 | 82.16 | 77.93 | 84.91 |
| $\alpha = 1.5, \epsilon = 0.8$ | 88.87 | 87.63 | 85.57 | 81.97 | **79.33** | 84.68 |
| *Step 1000 (4B tokens)* | | | | | | |
| $\alpha = 2.5, \epsilon = 0.2$ | **91.75** | 89.26 | 87.09 | 83.65 | 78.89 | 86.13 |
| $\alpha = 2.0, \epsilon = 0.4$ | 91.50 | **90.11** | **88.95** | **85.04** | **81.26** | **87.37** |
| $\alpha = 1.5, \epsilon = 0.8$ | 91.27 | 89.63 | 87.02 | 83.99 | 80.55 | 86.49 |

Table 15: Performance comparison between sequence-based and front-loaded supplementary context insertion across different context lengths (RULER scores).

| Method | 8k | 16k | 32k | 64k | 128k |
|---|---|---|---|---|---|
| Sequence | **92.00** | **91.17** | 85.10 | 83.26 | 78.25 |
| Front | 91.69 | 88.33 | **86.42** | **84.14** | **79.75** |

This comparison highlights the trade-off between $\alpha$ and $\epsilon$: both extremely loose and extremely strict thresholds (high $\alpha$ with low $\epsilon$, or low $\alpha$ with high $\epsilon$) eventually converge to similar effectiveness, while the balanced setting ($\alpha = 2.0, \epsilon = 0.4$) achieves the best overall performance under a comparable token budget.

## J EFFECT OF SUPPLEMENTARY CONTEXT INSERTION POSITION

To examine how the insertion position of supplementary documents affects performance, we conduct controlled experiments on synthetic long-context sequences. For each root document, we first construct a 64K-token sequence by keeping at most the top two entropy-reducing supplementary documents for each dependency and concatenating them with the root document. We then concatenate two such 64K sequences to form a 128K context containing two root documents and their corresponding supplementary contexts.

We compare two concatenation strategies:

(1) **Sequence**: for each root document, we build two 64K samples via our shuffle strategy and concatenate them sequentially:

[supp_context_1_shuffle_strategy] [root_doc1] [supp_context_2_shuffle_strategy] [root_doc2].

(2) **Front**: we shuffle all supplementary contexts corresponding to the two root documents and place them at the beginning, followed by the two root documents.:

[supp_context_1_2_shuffle_strategy] [root_doc1] [root_doc2].

As shown in Table 15, the trend suggests a shift in what helps the model depending on context length. In shorter ranges (8K–16K), keeping supplementary context close to its corresponding root document (*sequence*) provides clearer local cues and yields stronger performance. In longer ranges (32K–128K), front-loading the supplementary information (*front*) allows the model to better aggregate globally relevant signals over long spans, alleviating the difficulty of attending to distant supporting content.

Table 16: LLM-based categorization of relationships between verified contexts and high-entropy tokens (1,036 items).

| Category | #Items | Proportion |
|---|---|---|
| Positive signal (overall) | 762 | 73.55% |
|     Semantic elaboration | 268 | 25.87% |
|     Topical consistency | 256 | 24.71% |
|     Contextual background | 146 | 14.09% |
|     Factual support | 84 | 8.11% |
|     Narrative continuation | 6 | 0.58% |
|     Syntactic linking | 2 | 0.19% |
| Negative signal (overall) | 274 | 26.45% |
|     Unrelated | 265 | 25.58% |
|     Contradictory | 9 | 0.87% |

Table 17: Annotation of the primary level at which entropy is reduced for verified high-entropy tokens.

| Reduction Level | #Items | Proportion |
|---|---|---|
| Semantic-level reduction | 717 | 71.70% |
| Lexical-level reduction | 257 | 25.70% |
| Syntactic-level reduction | 26 | 2.60% |

## K   LLM-BASED ANALYSIS OF DEPENDENCY RELATIONSHIPS

To further understand the nature of the dependencies constructed by EntropyLong, we perform an LLM-based qualitative analysis of the relationship between verified contexts and their associated high-entropy tokens. We sample 1,036 high-entropy positions and their verified contexts and use Gemini 2.5 Flash (Comanici et al., 2025) to annotate which type of relationship is expressed. The specific prompt used for annotation is shown in Figure 5.

Table 16 summarizes the annotation statistics. Overall, 73.55% of cases are judged as positive signals, indicating that the verified contexts meaningfully support prediction at the high-entropy positions. Within these, the most common patterns are **Semantic Elaboration** (25.87%), where the context expands on or refines the local meaning, **Topical Consistency** (24.71%), where the context maintains the same topic and reinforces coherence, and **Contextual Background** (14.09%), where broader background or legal/historical information is supplied. The remaining positive cases include **Factual Support** (8.11%), **Narrative Continuation** (0.58%), and **Syntactic Linking** (0.19%). Negative cases (26.45%) are mostly labeled as **Unrelated** (25.58%), with a small fraction **Contradictory** (0.87%), illustrating that the entropy-based verification is not perfect but strongly biases toward useful dependencies.

For verified high-entropy tokens, we further ask the LLM to determine the level at which entropy reduction primarily occurs. As shown in Table 17, 71.70% of cases are labeled as **Semantic-level Reduction**, where the context clarifies meaning, fills in implicit premises, or disambiguates legal or factual references. 25.70% of cases are **Lexical-level Reduction**, where the context narrows down the plausible next words without substantially changing the underlying semantics, and 2.60% are **Syntactic-level Reduction**, where the context mainly constrains grammatical structure. These results suggest that EntropyLong predominantly constructs *semantic* rather than purely lexical or syntactic dependencies.

We also analyze high-entropy tokens for which verification failed, meaning no retrieved context passed the entropy-reduction threshold. For these cases, Gemini 2.5 Flash assigns qualitative causes such as **Inherent Diversity (Open Choice)** (51.10%), where multiple continuations remain equally plausible even with additional information, **Transitional or Structural Boundary** (36.15%), where high entropy reflects discourse or section transitions rather than missing knowledge, and **Information Gap** (12.75%), where the retrieved supplementary context did not pass the entropy-reduction threshold. These result show that many unverified positions arise from genuine ambiguity or structural factors rather than retrieval failure.

**LLM Prompt for Relationship Classification**

You are an expert linguistic and semantic analyzer.
Classify the relationship between:
(1) The original document segment
(2) The retrieved supplementary information
Choose ONE category from:
1. Factual Support
The retrieved content provides factual information (entities, events, numbers, definitions) that directly fills or clarifies the missing information at the high-entropy position.
2. Narrative Continuation
The retrieved content continues or supports the same narrative flow, event sequence, or storyline as the high-entropy context.
3. Syntactic Linking
The retrieved content provides structural or grammatical patterns that clarify the syntactic role needed at the high-entropy position.
4. Semantic Elaboration
The retrieved content offers definitional, descriptive, or semantic expansion related to the concept near the high-entropy position.
5. Contextual Background
The retrieved content adds broader background or contextual information (historical, domain-specific, situational) relevant to understanding the high-entropy position.
6. Topical Consistency
The retrieved content is topically related to the high-entropy context but does not provide direct semantic, narrative, or factual support.
7. Contradictory
The retrieved content conflicts with or contradicts the meaning or factual information implied by the high-entropy context.
8. Unrelated
The retrieved content is irrelevant to the high-entropy context with no topical, semantic, narrative, or syntactic connection.
Original document: {origin_text}
High-entropy word: {word}
Sentence context: {sentence}
Retrieved supplementary information: {retrieved_text}
Only output the relationship type.

Figure 5: Prompt used for LLM-based classification of relationships between verified contexts and high-entropy tokens.

> **Original Document ($D$):**
> ...Starting from a suggestion of Stephen Toulmin and through an interpretation of Neurath's (one of the founders of the Vienna Circle) **criticism** of Descartes' views on science, the paper attempts to outline a pattern of modernity opposed to the Cartesian one, that has been obtaining over the last four centuries....
>
> **Model State at "criticism":**
> High Entropy: The model is unsure of what specific aspect of Descartes' views Neurath criticized. It could be "methodology," "epistemology," "metaphysics," "dualism," etc.
>
> **Retrieved & Verified Context ($D'$):**
> ...Neurath disputed the existence of universal truths or natural laws. Decisions would never cease toentail a measure of uncertainty and men would always err in the forest of Descartes, without anyhope of ever exiting it. Rationality limited the pain of deciding exactly as other methods did andcould not claim to be the way out of the forest. Rationality, though, even if its results did not standthe test of truth...
>
> **Model State with Prepended Context ($[D'; D]$):**
> Low Entropy at "criticism": The model can now confidently predict "criticism" based on the retrieved context showing Neurath's opposition to Cartesian rationalism and universal truths.
> Top Prediction: {'criticism'} $\rightarrow$ completing the phrase "Neurath's criticism of Descartes' views on science"

Figure 6: An example of contextual uncertainty resolution. Retrieved context about Neurath's philosophical stance (disputing universal truths and Cartesian rationality) enables the model to confidently predict "criticism" by connecting this background knowledge with the original document's reference to Neurath's position toward Descartes' views on science.

## L    ADDITIONAL DISCUSSION: RELATION TO RETRIEVAL-BASED METHODS

Nearest-neighbor language models (Khandelwal et al., 2019) and Infinigram (Liu et al., 2024b) both operate at inference time: they retrieve text fragments that are semantically or statistically similar to the current context in order to reduce predictive entropy and directly improve next-token prediction. In contrast, EntropyLong uses entropy reduction as a proxy signal during pretraining to construct and filter training samples, thereby enabling the model to effectively learn long-range dependencies over extremely long contexts in the course of training.

## M    QUALITATIVE ANALYSIS: LONG-RANGE DEPENDENCY CONSTRUCTION

To illustrate how EntropyLong constructs meaningful long-range dependencies, we present a qualitative example showing how retrieved contextual information helps the model establish connections across extended contexts. As shown in Figure 6, our method identifies uncertainty positions and provides relevant background knowledge that enables better understanding of distant relationships in the text.

**Contextual Knowledge Integration Analysis.** This example demonstrates three critical aspects of EntropyLong's approach: (1) **Uncertainty Detection**: The model correctly identifies "criticism" as a high-entropy position requiring external context, (2) **Relevant Context Retrieval**: The system successfully retrieves Neurath's philosophical stance on Cartesian rationalism, which provides the necessary conceptual framework, and (3) **Synthesis Verification**: The retrieved context demonstrably reduces entropy by enabling the model to connect Neurath's opposition to universal truths and Cartesian rationality with the specific critique mentioned in the original document.

This example illustrates how EntropyLong constructs training instances that require genuine multi-step reasoning rather than simple pattern matching. The model must perform conceptual reasoning (understanding Neurath's philosophical position against Cartesian views) and contextual inference (connecting this philosophical stance to the specific criticism mentioned) to resolve the uncertainty.

Such complex dependencies are precisely what enable effective long-context understanding in real-world scenarios.

## N  AI ASSISTANCE DISCLOSURE

We acknowledge the use of large language models (LLMs) to assist with language polishing and grammar checking during the writing process. All core ideas, methodology, experimental design, and scientific contributions are entirely the work of the human authors. The LLM assistance was limited to improving readability and correcting grammatical errors, without contributing to the conceptual or technical content of this work.

