# OpenReview forum: "EntropyLong: Effective Long-Context Training via Predictive Uncertainty"
_ICLR.cc/2026/Conference — ICLR 2026 Poster_

### Official Review · Reviewer_L2EC · 2025-10-24

**Soundness:** 4
**Presentation:** 4
**Contribution:** 3
**Rating:** 6
**Confidence:** 5

**Summary:**

This paper tackles the lack of genuine long-range dependencies in long-context training data. It proposes EntropyLong, a novel data pipeline that uses a "model-in-the-loop" to verify data quality. The method identifies high-uncertainty (high-entropy) token positions in a document, retrieves relevant context, and verifies that this context reduces the model's prediction entropy at that position. Only verified context is used to construct final training samples. Training a Llama-3-8B model on this data shows significant performance gains on RULER and LongBench-v2 benchmarks over strong baselines, with ablations confirming the necessity of the entropy verification step.

**Strengths:**

1. The authors proposed a novel idea to create true long-context dependencies in long-context pre-training data. Their method using entropy to select tokens and relevant contexts shows effective results to answer the premise that a good dependency is one that measurably reduces model uncertainty.
2. This paper demonstrates clear and consistent performance improvements over two baselines (Quest and NExtLong) on RULER and LongBenchv2 benchmarks.
3. The analysis in section 6 is solid and convincing, providing detailed hyper-parameter selection and validating the paper's core hypothesis.

**Weaknesses:**

1. The proposed method needs a series of ablation studies to find effective thresholds. If we want to apply to a new model, we need to carefully find high entropy tokens and select dependent contexts. I'd like to see if this dataset can be generalized to other models besides the one (Llama3-8B) used to construct the dataset.
2. The baselines and the proposed method focus on finding extra documents for each short chunk to create long dependencies. We may need additional baselines by creating long dependencies based on current chunks like CIP[1] or UTK[2].
3. In section 3.2 item 2, the authors claim the optimal context C is the one that maximizes the information gain. However, in table 5, we saw the performance doesn't improve if we pick highest information gain contexts. The authors believe the problem is because of the data quantity. Then can we ensure the total tokens are the same with higher threshold to validate this claim?

[1] CIP: LongSkywork: A Training Recipe for Efficiently Extending Context Length in Large Language Models
[2] UTK: Untie the Knots: An Efficient Data Augmentation Strategy for Long-Context Pre-Training in Language Models

**Questions:**

1. For each document, you will find prepended contexts by changing two thresholds including high-entropy tokens and effective contexts to reduce entropy. How did you maintain the size of total length to be 128K in your case? If the contexts + original document > 128K, will you just separate as different samples like <context1><doc> and <context2><doc>? If so, does this mean we may train more on the tokens in original document <doc> than other documents?
2. The proposed threshold will affect the quality and quantity of the data. I think this work can be stronger if we analyze the trade-off between $\alpha$ and $\epsilon$ to have similar total token budget but one is high $\alpha$ and low $\epsilon$ and the other is low $\alpha$ and high $\epsilon$.

---

> ### Author Response · Authors · 2025-11-20
>
> **Questions 1-1.** For each document, you will find prepended contexts by changing two thresholds including high-entropy tokens and effective contexts to reduce entropy. How did you maintain the size of total length to be 128K in your case?
>
> **Answer 1-1:**
>
> After selecting dependency positions under different thresholds, we obtain a set of verified supplementary context segments:$supp1\text{-}1, supp1\text{-}2, \dots, supp5\text{-}4, \dots, supp x\text{-}z$, along with their corresponding root documents. To strictly maintain a total sample length of 128K tokens, we follow these steps:
>
> (1) Compute the remaining space available for supplementary context. For each root document, we calculate:$budget=128×1024−len(root\_document)$, this budget represents the maximum total length available for all supplementary context segments.
>
> (2) Sort all supplementary context segments by entropy reduction. We rank all candidate segments in descending order according to their entropy-reduction value at the target high-entropy positions, so that the most informative segments are added first.
>
> (3) Fill the supplementary context space segment by segment. We iterate through the sorted list and add segments one by one, updating the remaining space after each addition. If adding the next segment would exceed the budget, we stop filling and discard the remaining segments, ensuring that the final sample does not exceed 128K tokens.
>
> Discard samples that are too short. If the total length of “all available supplementary context + root document” is still below 128K tokens (i.e., not long enough to form a sufficiently long training instance), we discard the sample entirely and do not include it in the training set.
>
> **Questions 1-2.** does this mean we may train more on the tokens in original document [object Object] than other documents?
>
> **Answer 1-2:**
>
> No, each original document(root document) only generates one training sample
>
> **Questions 2.** The proposed threshold will affect the quality and quantity of the data. I think this work can be stronger if we analyze the trade-off between and $\epsilon$ to have similar total token budget but one is high $\alpha$ and low $\epsilon$and the other is low $\alpha$ and high $\epsilon$.
>
> **Answer 2:**
>
> A higher high-entropy selection threshold ($\alpha$ = 2.5) combined with a lower entropy-reduction verification threshold ($\epsilon$ = 0.2) allows the model to learn long-range dependencies quickly. However, this configuration tends to plateau and does not continue improving with additional training. In contrast, the configurations ($\alpha$ = 2.0, $\epsilon$ = 0.4) and ($\alpha$ = 1.5, $\epsilon$ = 0.8) produce similar improvement curves during training, but the threshold combination we ultimately adopt yields consistently better final performance.
>
> This comparison highlights the trade-off between $\alpha$ and $\epsilon$: both extremely loose and extremely strict thresholds converge to similar effectiveness, while a balanced setting ($\alpha$ = 2.0, ε = 0.4) provides the best overall performance under a similar token budget.
>
> Step 200 (0.8B tokens)
> | Setting          | 8K        | 16K       | 32K       | 64K       | 128K      | Avg       |
> | ---------------- | --------- | --------- | --------- | --------- | --------- | --------- |
> | α = 2.5, ε = 0.2 | 90.58     | 88.95     | **85.43** | **80.94** | **78.74** | **84.93** |
> | α = 2.0, ε = 0.4 | 90.90     | 87.03     | 83.13     | 77.82     | 72.99     | 82.37     |
> | α = 1.5, ε = 0.8 | **91.33** | **89.57** | 82.75     | 79.22     | 70.13     | 82.60     |
>
> Step 600 (2.4B tokens)
> | Setting          | 8K        | 16K       | 32K       | 64K       | 128K      | Avg       |
> | ---------------- | --------- | --------- | --------- | --------- | --------- | --------- |
> | α = 2.5, ε = 0.2 | **91.20** | **89.71** | **87.13** | **84.46** | **79.80** | **86.46** |
> | α = 2.0, ε = 0.4 | 90.73     | 88.08     | 85.63     | 82.16     | 77.93     | 84.91     |
> | α = 1.5, ε = 0.8 | 88.87     | 87.63     | 85.57     | 81.97     | 79.33     | 84.68     |
>
> Step 1000 (4B tokens)
> | Setting          | 8K        | 16K       | 32K       | 64K       | 128K      | Avg       |
> | ---------------- | --------- | --------- | --------- | --------- | --------- | --------- |
> | α = 2.5, ε = 0.2 | **91.75** | 89.26     | 87.09     | 83.65     | 78.89     | 86.13     |
> | α = 2.0, ε = 0.4 | 91.50     | **90.11** | **88.95** | **85.04** | **81.26** | **87.37** |
> | α = 1.5, ε = 0.8 | 91.27     | 89.63     | 87.02     | 83.99     | 80.55     | 86.49     |

---

> ### Author Response · Authors · 2025-11-20
>
> **Weaknesses 1.** The proposed method needs a series of ablation studies to find effective thresholds. If we want to apply to a new model, we need to carefully find high entropy tokens and select dependent contexts. I'd like to see if this dataset can be generalized to other models besides the one (Llama3-8B) used to construct the dataset.
>
> **Answer W1:**
>
> We expect that the general entropy distribution patterns identified by our method will be beneficial across different architectures. However, we acknowledge that each model may have its own optimal entropy thresholds due to differences in scale and the heterogeneity of pre-training corpora.
>
> To evaluate cross-model applicability, we applied the same entropy-based data construction settings to two additional models, Llama-3-8B-Instruct and Llama-3.2-3B, and compared their performance against NExtLong. The results in the tables below show that EntropyLong consistently outperforms NExtLong across all context lengths, demonstrating the strong transferability of the proposed approach.”
>
> Llama-3-8B-Instruct
> | method      | 8k    | 16k   | 32k   | 64k   | 128k  | avg   |
> | ----------- | ----- | ----- | ----- | ----- | ----- | ----- |
> | entropylong | 93.50 | 92.43 | 88.91 | 85.07 | 82.95 | 88.57 |
> | nextlong    | 92.39 | 90.41 | 87.72 | 82.69 | 80.96 | 86.84 |
>
>
> Llama-3.2-3B
> | method      | 8k    | 16k   | 32k   | 64k   | 128k  | avg   |
> | ----------- | ----- | ----- | ----- | ----- | ----- | ----- |
> | entropylong | 80.43 | 78.83 | 73.71 | 70.78 | 64.12 | 73.57 |
> | nextlong    | 79.33 | 77.98 | 71.69 | 70.34 | 62.35 | 72.34 |
>
>
> **Weaknesses 2.** The baselines and the proposed method focus on finding extra documents for each short chunk to create long dependencies. We may need additional baselines by creating long dependencies based on current chunks like CIP or UTK.
>
> **Answer W2:**
>
> We reconstructed the dataset following the UTK procedure and trained it for 4B tokens for a fair comparison. The results show that EntropyLong outperforms UTK across all context lengths on the RULER benchmark.
>
> | Method          | 8K        | 16K       | 32K       | 64K       | 128K      | Avg       |
> | --------------- | --------- | --------- | --------- | --------- | --------- | --------- |
> | UTK             | 89.51     | 84.66     | 80.23     | 77.84     | 71.98     | 80.84     |
> | **EntropyLong** | **91.50** | **90.11** | **88.95** | **85.04** | **81.26** | **87.37** |
>
> **Weaknesses 3.** In section 3.2 item 2, the authors claim the optimal context C is the one that maximizes the information gain. However, in table 5, we saw the performance doesn't improve if we pick highest information gain contexts. The authors believe the problem is because of the data quantity. Then can we ensure the total tokens are the same with higher threshold to validate this claim?
>
> **Answer W3:**
>
> We increased the threshold to obtain a more strictly filtered dataset, while ensuring that the verified dependency relations remained comparable to those identified under the original configuration. The resulting subset contained approximately 2.4B tokens. We subsequently trained a model on this higher-threshold subset, and the results are presented in the table below. Using a higher verification threshold (σ = 0.6) yields greater performance gains at longer context lengths.
>
> | Method     | 8K       | 16K      | 32K      | 64K      | 128K     | Avg      |
> |------------|----------|----------|----------|----------|----------|----------|
> | σ = 0.4    | **90.73** | 88.08    | **85.63** | 82.16    | 77.93    | 84.91    |
> | σ = 0.6    | 89.48    | **88.16** | 84.91    | **83.05** | **80.82** | **85.18** |
>
> The results show that with a higher verification threshold, selecting contexts that maximize information gain indeed improves performance.

---

> > ### Comment · Reviewer_L2EC · 2025-11-26
> >
> > Thanks authors for the thorough and detailed response. Most of my questions and weakness has been addressed. The additional ablation experiments including selection and verification threshold trade-off and higher verification threshold under similar verified numbers of tokens have strengthen the contribution. However,  I would like to see whether your proposed dataset has a broader contribution and can be used to other model families. You have showed the dataset can be generalized to other llama models. Can we try on other models except Llama? Or maybe you think the efforts to create a new dataset for each model is negligible. If so, can you show how much cost (compute and time) do we need to create a dataset?

---

> > > ### Author Response · Authors · 2025-11-27
> > >
> > > **1.**  You have showed the dataset can be generalized to other llama models. Can we try on other models except Llama?
> > >
> > > **Answer1:**
> > >
> > > To validate whether our proposed dataset provides broader benefits beyond the Llama model family, we further conduct experiments on Qwen2.5-3B, which is a pretrained model with a native context length of 32k and is extended to 128k in our setting. Specifically, we train Qwen2.5-3B using the dataset generated by Llama3-8B and report its performance at 32k, 64k, and 128k, comparing it against NExtLong.
> > >
> > > | method      | 32k    | 64k    | 128k   |
> > > |-------------|--------|--------|--------|
> > > | NExtLong    | **72.44** | 61.96  | 56.37  |
> > > | EntropyLong | 71.32  | **67.04** | **63.03** |
> > >
> > > These results show that the dataset consistently improves long-range dependency modeling, demonstrating that it can generalize to models beyond the llama family.
> > >
> > >
> > > **2.** Or maybe you think the efforts to create a new dataset for each model is negligible. If so, can you show how much cost (compute and time) do we need to create a dataset?
> > >
> > > **Answer2:**
> > >
> > > Constructing a new dataset on another model involves two main components: (1) identifying high-entropy positions and (2) performing verified-document validation. In our configuration, identifying high-entropy positions requires approximately 3 GPU hours, while verified-document validation requires about 352 GPU hours (using an 8B model to construct 4B tokens of data, as shown in Appendix E). Overall, this accounts for 20.93% of the total cost of dataset construction plus model training.

---

### Official Review · Reviewer_PDbx · 2025-10-27

**Soundness:** 3
**Presentation:** 3
**Contribution:** 2
**Rating:** 4
**Confidence:** 4

**Summary:**

In this paper, the authors introduce a methodology for improving the long-context performance of language models by modifying training data based on high-entropy regions. The approach begins by identifying positions in a training document where the model shows high predictive uncertainty, measured through token-level entropy. For each such high-entropy region, semantically similar documents are retrieved using embedding similarity. Among the retrieved candidates, only those that, when prepended to the original document, lead to a measurable reduction in entropy for that region are kept. These verified documents are then added to the beginning of the training document, producing an extended version that contains additional context proven to lower uncertainty.

The modified corpus is then used to fine-tune a pretrained model with RoPE-scaled positional embeddings to support longer context lengths. Experimental results show clear gains on long-context benchmarks such as RULER and LongBench-v2, confirming that entropy-guided data construction improves a model’s ability to utilize distant context. The paper also includes ablations on the entropy threshold, the necessity of the verification step, and the effect of the retrieval window size.

**Strengths:**

The experimental setup in this paper is mostly clear, and the idea of using uncertainty in text and explicitly targeting those regions is interesting. The core motivation—to improve model performance by providing additional, helpful information in the training text—is well-founded. By doing so, the approach encourages the model to make better use of context and, directly or indirectly, to rely on information that is placed farther away from the main document, thereby improving its ability to utilize long-range context. Overall, the idea opens the door to several interesting research directions that explore uncertainty-guided data modification and its role in enhancing context understanding in language models.

**Weaknesses:**

- **W1)** One weakness is the lack of clarity in how high-entropy regions are identified and represented. The paper mentions detecting areas of high predictive entropy, but it is unclear how these local and often volatile entropy values are aggregated into meaningful regions that actually require additional context. In practice, entropy can fluctuate sharply even within short spans—for instance, the first part of a word or phrase often has higher entropy than later parts due to language-level artifacts. Without clear illustrations or examples of how such regions behave during next-token prediction, it is difficult to assess whether they truly capture model uncertainty rather than incidental variation.

- **W2)** Conceptually, the approach still resembles retrieval augmentation, where related text is added to provide more context, and it is not entirely clear whether entropy is the main factor behind the gains. Although the authors compare verified versus unverified document additions, the difference could stem from how the top-k similar documents are chosen or from context-overload effects. It would help to clarify whether documents that reduce entropy tend to rank higher in similarity, and how many are included when verification is skipped. While EntropyLong operates at a finer granularity than RE³SYN, a direct comparison would strengthen the argument that entropy reduction—rather than dependency linking alone—is what drives improvement.

- **W3)**  The paper largely treats high entropy as a sign of poor model performance or uncertainty, but this link is not always straightforward. This claim is somewhat reflected by the language of this paper, specially cases where the authors claim this entropy increase as a theory for poor model performance. Entropy in language can also reflect diversity or ambiguity rather than failure, and its interpretation varies across models and data domains. It would help if the authors discussed whether lower entropy consistently aligns with better predictions or cited supporting evidence for that assumption. From my own experience with large language models, entropy behavior can differ substantially across model families and scales, suggesting that the same thresholds for $\alpha$ and $\epsilon$ may not generalize. Including experiments that vary these parameters across different models would clarify how model-dependent the method is. I understand that the selection of such values is document dependant, but I believe the model itself could play a role in this method which explicitly uses entropy  of a language model for targetting points of improvement.

- **W4)** The paper does not provide information or comparisons regarding the computational cost of the entropy-based data curation process. I assume that the lookup and top-k extraction steps could be computationally intensive, and it would be helpful if the authors included at least an approximate cost analysis or comparison with other data construction methods, even if only in the appendix.

- **W5)**  (Minor) The paper does not discuss several closely related approaches that also reuse similar regions of text to improve next-token prediction. Khandelwal et al. (2020, Nearest Neighbor Language Models) retrieve semantically similar examples from the training set using embedding similarity between hidden states, improving predictions by grounding generation in past contexts. In contrast, Liu et al. (2025, Infinigram) rely on n-gram frequency statistics to reuse highly similar text fragments during inference. While these methods apply retrieval at inference rather than training, both demonstrate that referencing similar parts of text can enhance next-token accuracy. Highlighting this connection would make it clearer how EntropyLong extends these retrieval-style ideas into a training-time, entropy-guided framework.

**Questions:**

- **Q1**: I am curious about the placement and ordering of the retrieved, entropy-reducing documents. Table 7 suggests that random or sequential shuffling sometimes yields noticeable gains—especially at 32 K and 128 K context lengths—which might indicate sensitivity to document ordering. Do the authors have an explanation for this behavior?
Additionally, connecting to works such as Lost in the Middle (Liu et al., 2023) and LDAM (Chen et al 2025), which study how the position of relevant information affects model performance, it would be interesting to see experiments that vary where these documents are inserted. My intuition is that placing them at the beginning helps the model learn to attend to earlier parts of the context, potentially explaining the stronger performance at longer lengths.

- **Q2**: It is somewhat unclear why the authors choose to append multiple retrieved documents rather than focusing on a single or very limited number that produce the largest entropy reduction. Would using only the top document (K = 1) that most strongly reduces entropy yield similar improvements? Exploring smaller K values could help disentangle the effects of information quantity from information placement, and—when combined with varying document position—might offer clearer insights into how LLMs actually utilize extended context.

- **Q3**: I am having a hard time understanding Figure 2 from description and text. Could the authors please provide an explanation of the setup, formulation for the attention pattern analysis and how it relates to the long entropy method ?

---

> ### Author Response · Authors · 2025-11-20
>
> **Questions 1.** I am curious about the placement and ordering of the retrieved, entropy-reducing documents. Table 7 suggests that random or sequential shuffling sometimes yields noticeable gains—especially at 32 K and 128 K context lengths—which might indicate sensitivity to document ordering. Do the authors have an explanation for this behavior?
>
> Sequential shuffling helps prevent the model from learning a positional bias tied to the relative ordering of supplementary documents. As a result, the model is forced to search the entire context to locate the relevant evidence rather than relying on shortcut positional cues. This reliance on global retrieval becomes especially beneficial at long context lengths (e.g., 32K and 128K), which likely explains why the sequential variant yields noticeably larger gains in these settings.
>
> **Questions 2.** It is somewhat unclear why the authors choose to append multiple retrieved documents rather than focusing on a single or very limited number that produce the largest entropy reduction. Would using only the top document (K = 1) that most strongly reduces entropy yield similar improvements? Exploring smaller K values could help disentangle the effects of information quantity from information placement, and—when combined with varying document position—might offer clearer insights into how LLMs actually utilize extended context.
>
> Using only the supplementary document that yields the largest entropy reduction is not sufficient to fill the 128K sequence. Each 128K training sample contains only one root document, and the remaining content consists entirely of supplementary documents. We conducted an additional comparison experiment that combines smaller values of K with different placement positions of the supplementary documents to analyze how LLMs perform long-context extension. The final model performance is consistent with the reviewer’s intuition that placing supplementary documents at the beginning helps the model learn to attend to earlier parts of the context.
>
> To examine how the insertion position of supplementary documents affects performance, we designed two controlled comparison experiments. For each root document, we synthesized a 64K sequence (keeping at most the top two entropy-reducing supplementary documents for each dependency). We then concatenated two such 64K sequences to form a 128K context.
>
> **Setting1 (sequence)**:[supp_context_1_shuffle_strategy] [root_doc1] [supp_context_2_shuffle_strategy] [root_doc2]
>
> **Setting 2 (front)**: We place all supplementary contexts at the beginning and both root documents at the end:  [supp_context_1_2_shuffle_strategy] [root_doc1] [root_doc2]
>
> | Method    | 8K       | 16K      | 32K      | 64K      | 128K     |
> |-----------|----------|----------|----------|----------|----------|
> | **Sequence** | **92.0** | **91.17** | 85.1     | 83.26    | 78.25    |
> | Front     | 91.69    | 88.33    | **86.42** | **84.14** | **79.75** |
>
> The results for different insertion positions of supplementary documents are shown in the table above.
>
> Short ranges (8K–16K): Keeping supplementary context close to its corresponding root document (the sequence strategy) provides clearer local cues, leading to stronger performance.
>
> Long ranges (32K–128K): By front-loading the supplementary information, the model can more effectively learn globally relevant signals over long spans, thereby mitigating the difficulty of attending to distant supporting content.
>
>
> **Questions 3.**  I am having a hard time understanding Figure 2 from description and text. Could the authors please provide an explanation of the setup, formulation for the attention pattern analysis and how it relates to the long entropy method ?
>
> We designed this experiment to evaluate how models trained on datasets constructed using different methods perform on truly long-range dependencies. Specifically, we sampled 1,024 examples from the HotpotQA[1] dataset, where each example consists of three parts: the question, a text block containing the answer, and other similar but irrelevant text blocks.
>
> To study the model’s attention behavior, we computed the attention from the answer token to the text block containing the correct answer during inference. For comparison, we also evaluated the second-best method ( NExtLong ).
>
> In Figure~A, the “Context Number” represents the amount of irrelevant content (a higher Context Number indicates a longer context, with the answer-containing text block placed at the beginning).
>
> In Figure~B, we report the relative increase in attention from the answer token to the text block containing the correct answer compared with the nextlong method, computed as
> $\frac{A_{\text{EntropyLong}} - A_{\text{NextLong}}}{A_{\text{NextLong}}}$, with the number of context blocks fixed at 30.
>
>
> [1] Yang Z, Qi P, Zhang S, et al. HotpotQA: A dataset for diverse, explainable multi-hop question answering

---

> ### Author Response · Authors · 2025-11-20
>
> **Weaknesses 1.** One weakness is the lack of clarity in how high-entropy regions are identified and represented. The paper mentions detecting areas of high predictive entropy, but it is unclear how these local and often volatile entropy values are aggregated into meaningful regions that actually require additional context. In practice, entropy can fluctuate sharply even within short spans—for instance, the first part of a word or phrase often has higher entropy than later parts due to language-level artifacts. Without clear illustrations or examples of how such regions behave during next-token prediction, it is difficult to assess whether they truly capture model uncertainty rather than incidental variation.
>
> When computing entropy, we perform inference over the entire document and calculate the entropy for each token. The term “region,” as we understand it, refers to the query span used to retrieve supplementary content. Specifically, for each high-entropy token, we take the 16 tokens before and after its position.
>
> We conducted a qualitative analysis with Gemini2.5 Flash on the verified regions together with their retrieved supplementary content, and we also analyzed the high-entropy positions that did not pass verification. The analysis shows that 73.55% of the verified high-entropy regions exhibit strong dependency relations with the supplementary content (the top-1 entropy-reducing text). Among the high-entropy positions that were not verified, 51.1% stem from inherent linguistic variability that cannot be reduced, 25.7% arise from entropy spikes at transitional boundaries, and the remaining 12.75% correspond to genuine information gaps. The detailed analysis is provided below.
>
> | Category          | Count | Percentage (%) |
> |----------------------------|-------|----------------|
> | **Positive (73.55%)** |       |                |
> | Semantic Elaboration       | 268   | 25.87          |
> | Topical Consistency        | 256   | 24.71          |
> | Contextual Background      | 146   | 14.09          |
> | Factual Support            | 84    | 8.11           |
> | Narrative Continuation     | 6     | 0.58           |
> | Syntactic Linking          | 2     | 0.19           |
> | **Negative (26.45%)** |       |                |
> | Unrelated                  | 265   | 25.58          |
> | Contradictory              | 9     | 0.87           |
> | **Total Items**            | 1036  | 100.00         |
>
> Below is a case classified as a Contextual Background relationship:
>
> **sentence information:**
>
> ...seearts. 6 and 9 of the 1885 Berlin Act. See confirmation of the concept of ‘crimes against humanity’ in UNGA Res 3 (I) (Feb. 13, 1946); UNGA Res 95 (I) (Dec. 11, 1946)...
>
> **retrivaled text:**
>
> ... pass and allowed them to continue their ongoing forms of genocide and other crimes against humanity. Rather than take a principled stand in favour of all victims of genocide, Lemkin picked his battles, and he sided with the victors of the Second World War. These countries, in his view, would constitute the international champions of the convention in preventing genocide abroad, but not at home. This myopic view of history created several problems. First, in helping to draft the convention, Western settler states actively excluded elements that could be compromising in their own domestic contexts. Second, in supporting the convention during its evolution, they gained rhetorical assurances from Lemkin that their adversaries were genocidal while they were not. Third, in promoting a high level of respect for the sovereignty of states, states could effectively shield themselves domestically by choosing how they would incorporate or not incorporate the convention into their own legislation...
>
>
> **Weaknesses 2.** Conceptually, the approach still resembles retrieval augmentation, where related text is added to provide more context, and it is not entirely clear whether entropy is the main factor behind the gains. Although the authors compare verified versus unverified document additions, the difference could stem from how the top-k similar documents are chosen or from context-overload effects. It would help to clarify whether documents that reduce entropy tend to rank higher in similarity, and how many are included when verification is skipped. While EntropyLong operates at a finer granularity than RE³SYN, a direct comparison would strengthen the argument that entropy reduction—rather than dependency linking alone—is what drives improvement.
>
> We examined the verified supplementary contexts and found that the average order of documents producing the largest entropy reduction was 18.85 (with top-k = 32), and the overlap between non-verified and verified contexts was 15.6%.
>
> RE³SYN has not released the training dataset, and due to time constraints, we were unable to reproduce their method.

---

> ### Author Response · Authors · 2025-11-20
>
> **Weaknesses 3-1.**  The paper largely treats high entropy as a sign of poor model performance or uncertainty, but this link is not always straightforward. This claim is somewhat reflected by the language of this paper, specially cases where the authors claim this entropy increase as a theory for poor model performance. Entropy in language can also reflect diversity or ambiguity rather than failure, and its interpretation varies across models and data domains. It would help if the authors discussed whether lower entropy consistently aligns with better predictions or cited supporting evidence for that assumption.
>
> We would like to clarify that we do not claim entropy increase as a theory for poor model performance. Throughout the paper, we use entropy only as an indicator of the model’s uncertainty at specific positions, not as a direct measure of prediction quality. Higher entropy simply reflects that the model is less confident about the next-token distribution; it does not imply that the model is performing poorly overall. Our method leverages these high-uncertainty positions to identify where additional context may be beneficial, rather than drawing conclusions about model performance.
>
> **Weaknesses 3-2.** From my own experience with large language models, entropy behavior can differ substantially across model families and scales, suggesting that the same thresholds for $\alpha$ and $\epsilon$ may not generalize. Including experiments that vary these parameters across different models would clarify how model-dependent the method is. I understand that the selection of such values is document dependant, but I believe the model itself could play a role in this method which explicitly uses entropy of a language model for targetting points of improvement.
>
> We agree with the reviewer’s perspective. Our method is inherently model-in-the-loop, meaning that the long-context data it produces is tailored to the specific model whose long-range dependencies are being strengthened. The parameters indeed play different roles:
>
>  $\alpha$ controls the number of initial high-entropy candidates considered for verification.
>
> $\epsilon$ affects how many positions remain after the verification step.
>
> To examine cross-model behavior, we compared Qwen3-8B and Llama3-8B under the same entropy-filtering settings. Although the two models differ in architecture, their selected positions show a moderate degree of consistency:
>
> Among the α-selected high-entropy tokens, 74.81% appear in both models.
>
> After verification, 57.09% of the final positions overlap, and the number of verified positions differs only slightly.
>
> Due to time constraints, we were unable to further evaluate whether the synthesized dataset is also optimal for Qwen3-8B, nor to run a broader sweep of hyperparameters across multiple architectures. Thus, although the exact hyperparameter choices may not be optimal for every architecture, we expect that applying the same settings to other models would still yield strong performance.
>
> **Weaknesses 4.** The paper does not provide information or comparisons regarding the computational cost of the entropy-based data curation process. I assume that the lookup and top-k extraction steps could be computationally intensive, and it would be helpful if the authors included at least an approximate cost analysis or comparison with other data construction methods, even if only in the appendix.
>
> We appreciate the reviewer’s suggestion. In the revised version, we will include an approximate computational cost analysis of our entropy-based data curation pipeline, along with a comparison to baseline data construction methods, and place this discussion in the appendix.
>
> **Weaknesses 5.** The paper does not discuss several closely related approaches that also reuse similar regions of text to improve next-token prediction. Khandelwal et al. (2020, Nearest Neighbor Language Models) retrieve semantically similar examples from the training set using embedding similarity between hidden states, improving predictions by grounding generation in past contexts. In contrast, Liu et al. (2025, Infinigram) rely on n-gram frequency statistics to reuse highly similar text fragments during inference. While these methods apply retrieval at inference rather than training, both demonstrate that referencing similar parts of text can enhance next-token accuracy. Highlighting this connection would make it clearer how EntropyLong extends these retrieval-style ideas into a training-time, entropy-guided framework.
>
> We thank the reviewer for pointing out these closely related retrieval-based approaches. In the revised version, we will expand the Related Work section to include a discussion of Nearest Neighbor LMs (Khandelwal et al., 2020) and Infinigram (Liu et al., 2025), and clarify how our method connects to and differs from these inference-time retrieval techniques by extending the idea of leveraging similar text into a training-time, entropy-guided framework.

---

> > ### Author Response · Authors · 2025-11-21
> >
> > **Supplementary Analysis for Weakness 3-2**
> >
> > To further support this point, we also examined cross-model applicability by applying the same entropy-based data construction settings to two additional models, Llama-3-8B-Instruct and Llama-3.2-3B, and comparing their performance with the NExtLong baseline. The results in the tables below show that EntropyLong consistently outperforms nextlong across all context lengths on both models, demonstrating strong transferability of the proposed approach.
> >
> > Llama-3-8B-Instruct
> > | method      | 8k        | 16k       | 32k       | 64k   | 128k      | avg       |
> > | ----------- | --------- | --------- | --------- | ----- | --------- | --------- |
> > | entropylong | **93.50** | **92.43** | **88.91** | 85.07 | **82.95** | **88.57** |
> > | nextlong    | 92.39     | 90.41     | 87.72     | 82.69 | 80.96     | 86.84     |
> >
> > Llama-3.2-3B
> > | method      | 8k        | 16k       | 32k       | 64k       | 128k      | avg       |
> > | ----------- | --------- | --------- | --------- | --------- | --------- | --------- |
> > | entropylong | **80.43** | **78.83** | **73.71** | **70.78** | **64.12** | **73.57** |
> > | nextlong    | 79.33     | 77.98     | 71.69     | 70.34     | 62.35     | 72.34     |

---

> ### Comment · Reviewer_PDbx · 2025-11-22
>
> First I would like to thank the authors for their hard work and elaborate explanations. I understand the tie is very short for rebuttal to address some of these points. I appreciate the work.
>
> Regarding Weakness 1: Can the authors please elaborate how they get the categories ? I am not familiar with this categorization.
>
>
> Regarding Weakness 2: Thank you for doing the experiment. I understand the comment regarding RESYN. But for the largerst entropy reduction I am not sure I understand the 18.85 ? I'm not sure I understand because the number seems a bit large ? could the authors give a scale forthe entropy of the tokens before and after ?
>
> I think if the authors can provide me with some information on how much % wise the entropy is reduced for the high-entropy tokens, say showing a bar plot or distirbution of entropy reduction, compared to other tokens (which which weren't flagged as high entropy) it would convince me for ths weaknesses above.

---

> > ### Author Response · Authors · 2025-11-22
> >
> > **1.** Can the authors please elaborate how they get the categories ? I am not familiar with this categorization.
> >
> > **Answer1:**
> >
> > We use Gemini2.5 Flash to annotate the dependency relations between high-entropy regions and their retrieved supplementary content. This follows a widely adopted practice [1][2] of leveraging LLMs as automatic annotators for text classification and semantic labeling, which has been shown to be accurate and cost-effective in many settings.
> >
> > The eight categories in Table are not imported from an existing formal taxonomy, but are a lightweight annotation scheme we designed to capture how the retrieved supplementary text relates to a high‑entropy region. Concretely, we first defined eight intuitive relation types — Factual Support, Narrative Continuation, Syntactic Linking, Semantic Elaboration, Contextual Background, Topical Consistency, Contradictory, and Unrelated — which roughly correspond to common discourse and semantic relations (e.g., factual vs. topical support, narrow semantic elaboration vs. broad background, etc.).
> >
> > We then used Gemini2.5 Flash to automatically assign one of these labels to each pair of (i) original document segment and (ii) its retrieved supplementary content. The model was prompted as an “expert linguistic and semantic analyzer” and instructed to choose exactly one label from the above list, given (a) the original document, (b) the high‑entropy word and its sentence context, and (c) the retrieved passage. The exact prompt is:
> >
> > > You are an expert linguistic and semantic analyzer.
> > >
> > > Classify the relationship between:
> > >
> > > (1) The original document segment
> > >
> > > (2) The retrieved supplementary information
> > >
> > > Choose ONE category from:
> > >
> > > 1. Factual Support
> > >
> > >    The retrieved content provides factual information (entities, events, numbers, definitions) that directly fills or clarifies the missing information at the high-entropy position.
> > > 2. Narrative Continuation
> > >
> > >    The retrieved content continues or supports the same narrative flow, event sequence, or storyline as the high-entropy context.
> > > 3. Syntactic_Linking
> > >
> > >    The retrieved content provides structural or grammatical patterns that clarify the syntactic role needed at the high-entropy position.
> > > 4. Semantic_Elaboration
> > >
> > >    The retrieved content offers definitional, descriptive, or semantic expansion related to the concept near the high-entropy position.
> > > 5. Contextual_Background
> > >
> > >    The retrieved content adds broader background or contextual information (historical, domain-specific, situational) relevant to understanding the high-entropy position.
> > > 6. Topical_Consistency
> > >
> > >    The retrieved content is topically related to the high-entropy context but does not provide direct semantic, narrative, or factual support.
> > > 7. Contradictory
> > >
> > >     The retrieved content conflicts with or contradicts the meaning or factual information implied by the high-entropy context.
> > >
> > > 8. Unrelated
> > >
> > >     The retrieved content is irrelevant to the high-entropy context with no topical, semantic, narrative, or syntactic connection.
> > >
> > > Original document:
> > > {origin_text}
> > >
> > > High-entropy word: {word}
> > >
> > > Sentence context: {sentence}
> > >
> > > Retrieved supplementary information:
> > >
> > > {retrieved_text}
> > >
> > > Only output the relationship type.
> >
> > [1] Gilardi F, Alizadeh M, Kubli M. ChatGPT outperforms crowd workers for text-annotation tasks[J]. Proceedings of the National Academy of Sciences, 2023, 120(30): e2305016120.
> > [2] Kocmi T, Federmann C. Large language models are state-of-the-art evaluators of translation quality[J]. arXiv preprint arXiv:2302.14520, 2023.

---

> > > ### Author Response · Authors · 2025-11-22
> > >
> > > **2-1.** I understand the comment regarding RESYN. But for the largerst entropy reduction I am not sure I understand the 18.85 ? I'm not sure I understand because the number seems a bit large ?
> > >
> > > **Answer2-1:**
> > >
> > > Regarding the value 18.85, this number is not the entropy itself, but the average rank (within the top-32 retrieved documents) of the document that yields the largest entropy reduction after verification.
> > >
> > > We retrieve the top-32 candidate documents based on embedding similarity and sort them from most similar to least similar. For each document we run our verification step and compute its entropy reduction. The document that reduces entropy the most typically appears around rank 18.85 on average.
> > >
> > > Therefore, 18.85 does not indicate an unusually large entropy value. Instead, it simply shows that:The document providing the highest information gain is often not the top-ranked document by similarity.
> > >
> > > Although nearest-neighbor retrieval is effective at retrieving topically relevant and potentially useful information, the item that most reduces uncertainty is not always the one with the highest embedding similarity. This pattern is consistent with prior findings[3][4] that similarity-based retrieval and uncertainty-reducing information are not perfectly aligned.
> > >
> > > **2-2.** could the authors give a scale for the entropy of the tokens before and after ?
> > >
> > > **Answer2-2:**
> > >
> > > For verified high-entropy tokens, the average entropy before adding supplementary information is 5.62, and after incorporating the document that maximally reduces entropy, the average entropy decreases to 1.7. The average information gain ( $\frac{H_{\text{before}} - H_{\text{after}}}{H_{\text{before}}}$ ) is 0.68, as reported in Section 5.1 of the paper.
> > >
> > > For high-entropy tokens that were not verified, the average entropy before adding supplementary information is 4.4, and after using the document that maximally reduces entropy, the average entropy decreases slightly to 3.84.
> > >
> > > The detailed distribution of entropy reductions for both verified and unverified high-entropy tokens will be provided in the updated version of the paper.
> > >
> > >
> > > [3] Yao Z, Qi W, Pan L, et al. Seakr: Self-aware knowledge retrieval for adaptive retrieval augmented generation[C]//Proceedings of the 63rd Annual Meeting of the Association for Computational Linguistics (Volume 1: Long Papers). 2025: 27022-27043.
> > >
> > > [4] Jiang Y, Zhao S, Li J, et al. GainRAG: Preference Alignment in Retrieval-Augmented Generation through Gain Signal Synthesis[J]. arXiv preprint arXiv:2505.18710, 2025.

---

> > > > ### Comment · Reviewer_PDbx · 2025-11-26
> > > >
> > > > Efficiency:
> > > > I would like to thank the authors for their comprehensive response.
> > > > With regards to efficiency, I understand that this method does require more time for pre-processing of data, but yields more efficient data for training. I accept this argument.
> > > >
> > > > Discussion of Related Work:
> > > > Regarding the discussion on previous work, I believe the work requires a more in depth discussion on mentioned methods.
> > > >
> > > > Entropy Reduction:
> > > > Regarding measuring the entropy reduction, I appreciate the authors providing further experiments confirming that the entropy does go down, providing the numbers of before an after gave me a better picture, I suggest the authors include this statistics in the main body. I am more confident that the entropy reduction is helping improve model performance.
> > > >
> > > > Questions 1 and 2:
> > > > I am happy with the response to questions 1 and 2. It is clear to me that authors conducted a considerable amount of experiments and I appreciate the effort and I am happy with the results.
> > > >
> > > > Attention Analysis / LDAM:
> > > > For Q3 and the analysis done for attention, I would like to direct the reviewers towards LDAM introduced in Chen et al 2025. I believe their method and that of the authors tend to have similarities that should be discussed as they perform very similar analysis, but with attention based metric for the choice of documents. I understand that this is relatively recent work and thus don't expect the authors to have included it before.
> > > >
> > > > Single-Token Entropy Concern:
> > > > Finally I would like to point out my argument against single token entropy analysis. I beleive entropy on single tokens could be noisy and misleading and that terms like uncertainty make much more sense when considering groups of words and sentences. I would argue for a metric investigating high entropy sentences/sections by doing averaging across chunks of text, finding the highest sections and targetting those areas. I am not particularly convinced with the LLM based token categorization, however I appreciate the effort as it provides some useful insights regarding token categorization.
> > > >
> > > > Final Score:
> > > > As a result, I am increasing my score to a 6 and wish the authors good luck!

---

### Official Review · Reviewer_2y86 · 2025-10-28

**Soundness:** 3
**Presentation:** 4
**Contribution:** 4
**Rating:** 10
**Confidence:** 4

**Summary:**

This paper proposes EntropyLong, a novel data-construction recipe for the “long-context continual pre-training” stage of LLM development. Given base model M, training corpus D, and retrieval corpus R, the authors present a procedure consisting of 4 steps:
1. For each document from D, define high uncertainty tokens (the tokens the autoregressive model M is unsure about)
2. For each uncertain token:

2.a. Form a query by taking the surrounding context

2.b. Embed the query (jina-embeddings-v3)

2.c. Use this query to retrieve 32 documents from R using cosine similarity

3. Filter documents that reduce the uncertainty of the token (only keep C if it did not appear for this doc before)
4. Shuffle and merge filtered documents into a context

Then, starting with the M checkpoint (LLaMa3-8B) as the initialization, they continue its training with increased sequence length (8K to 128K) as per standard practice with modified RoPE hyperparameter.

The authors compare their method to baseline methods: Quest (just retrieve related documents) and NExtLong (retrieve related documents, but also unrelated documents for distraction)

**Strengths:**

The authors clearly explain their strategy, formulate and test relevant hypotheses.

**Weaknesses:**

The authors claim that dataset release is a key contribution, but no comparison with other datasets is presented in the experiments section. They only compared their method against other published methods.
One comparison that could have been done is the comparison with the dataset that was used for training LLaMa-3.1-8B, which has the same context length as the resulting model presented by this work. Another comparison could be made with LongMIT-128K.

**Questions:**

Here are several questions for the authors:
- How are 100K root docs sampled? Uniformly or using some other random strategy?
- Are the root documents included in the retrieval corpus?
- How is underflow or overflow in ConstructSequence handled?
- How is context deduplicated across root documents (sequences)? How do you know there is no retrieved document that is a part of every - single sequence you train on?

---

> ### Author Response · Authors · 2025-11-20
>
> **Questions 1.** How are 100K root docs sampled? Uniformly or using some other random strategy?
>
> **Answer 1:**
>
> The 100K root documents are uniformly sampled from the pretraining corpus.
>
> **Questions 2.** Are the root documents included in the retrieval corpus?
>
> **Answer 2:**
>
> No, the root documents are not included. The retrieval corpus is constructed by removing the root documents from the pretraining corpus.
>
> **Questions 3.** How is underflow or overflow in ConstructSequence handled?
>
> **Answer 3:**
>
> After selecting dependency positions under different thresholds, we obtain a set of verified supplementary context segments:$supp1\text{-}1, supp1\text{-}2, \dots, supp5\text{-}4, \dots, supp x\text{-}z$, along with their corresponding root documents. To strictly maintain a total sample length of 128K tokens, we follow these steps:
>
> (1) Compute the remaining space available for supplementary context. For each root document, we calculate:$budget=128×1024−len(root\_document)$, this budget represents the maximum total length available for all supplementary context segments.
>
> (2) Sort all supplementary context segments by entropy reduction. We rank all candidate segments in descending order according to their entropy-reduction value at the target high-entropy positions, so that the most informative segments are added first.
>
> (3) Fill the supplementary context space segment by segment. We iterate through the sorted list and add segments one by one, updating the remaining space after each addition. If adding the next segment would exceed the budget, we stop filling and discard the remaining segments, ensuring that the final sample does not exceed 128K tokens.
>
> Discard samples that are too short. If the total length of “all available supplementary context + root document” is still below 128K tokens (i.e., not long enough to form a sufficiently long training instance), we discard the sample entirely and do not include it in the training set.
>
> **Questions 4-1.** How is context deduplicated across root documents (sequences)?
>
> **Answer 4-1:**
>
> For each sample (one root document plus its supplementary documents), if a supplementary document has already appeared in previous supplementary documents, we discard the duplicate.
>
> **Questions 4-2.** How do you know there is no retrieved document that is a part of every - single sequence you train on?
>
> **Answer 4-2:**
>
> In practice, we did not explicitly enforce this. However, we sampled 1,000 examples and analyzed the usage of supplementary documents. On average, each document was used 1.14 times, indicating minimal repetition across sequences.
>
> **Weaknesses** The authors claim that dataset release is a key contribution, but no comparison with other datasets is presented in the experiments section. They only compared their method against other published methods. One comparison that could have been done is the comparison with the dataset that was used for training LLaMa-3.1-8B, which has the same context length as the resulting model presented by this work. Another comparison could be made with LongMIT-128K.
>
> **Answer W:**
>
> Our data synthesis is applied during the pretraining stage, so it cannot be directly compared with long SFT datasets (LongMIT-128K[1]). Additionally, the dataset used to extend LLaMA 3.1-8B to a 128K context length has not been released. However, we were able to obtain evaluation results on the RULER benchmark[2], as shown below:
>
> | Model                 | 32K      | 64K      | 128K     |
> |-----------------------|----------|----------|----------|
> | LLaMA3.1-8B-128K      | 87.4     | 84.7     | 77.0     |
> | **Entropy-Long**       | **88.95** | **85.04** | **81.26** |
>
> [1] Chen Z, Chen Q, Qin L, et al. What are the essential factors in crafting effective long context multi-hop instruction datasets? insights and best practices[C]//Proceedings of the 63rd Annual Meeting of the Association for Computational Linguistics (Volume 1: Long Papers). 2025: 27129-27151.
>
> [2] NVIDIA. (2024). RULER – README. GitHub. Retrieved November 20, 2025, from https://github.com/NVIDIA/RULER/blob/main/README.md

---

### Official Review · Reviewer_dtra · 2025-10-29

**Soundness:** 2
**Presentation:** 3
**Contribution:** 2
**Rating:** 2
**Confidence:** 3

**Summary:**

This paper proposes EntropyLong, a model-in-the-loop data construction framework for long-context language model training. It identifies high-entropy positions where the model is uncertain, retrieves semantically related contexts, and retains only those that empirically reduce prediction entropy, ensuring genuine long-range dependencies. Using Llama-3-8B with a 128K context window trained on FineWeb-Edu and Cosmopedia, EntropyLong achieves higher scores than Quest and NExtLong on RULER (87.4 avg) and LongBench-v2 (27.6 overall) while preserving short-text performance. Ablations confirm that entropy-based verification is essential and that optimal thresholds balance data quality and quantity, leading to improved long-context reasoning and mitigation of the “lost-in-the-middle” problem.

**Strengths:**

The paper presents a novel application for long-context data construction, offering a model-driven perspective on how to identify and verify informative dependencies across distant text segments.

Its originality lies in connecting information-theoretic uncertainty with practical dataset synthesis, moving beyond heuristic concatenation methods.

The paper is well-organized and clear, making a complex pipeline easy to follow.

**Weaknesses:**

1. The central assumption is that high predictive entropy indicates missing long-range information, which lacks strong theoretical grounding. Empirically, many high-entropy tokens correspond to discourse openings, transitions, or rare words rather than genuine dependency points. Thus, the “verified” concatenations may enhance topical smoothness rather than causal linkage.

2. The paper does not analyze what types of dependencies are actually captured after verification (e.g., factual consistency, narrative continuity, or syntactic linking). Without qualitative inspection or causal tracing, the claim of “true long-range reasoning” remains speculative.

3. The framework requires per-sample forward passes for entropy computation and re-verification, which substantially increases computational cost. The paper does not discuss efficiency trade-offs or scaling to larger corpora.

4. Although the paper frames its contribution as constructing long-context dependencies, the retrieval query for each high-entropy token is limited to a 16-token local window. This narrow context restricts semantic scope to near-sentence continuations rather than true cross-document or distant dependencies, effectively reducing the retrieval process to localized topical or lexical matching. As a result, the constructed samples may improve surface fluency but do not necessarily capture genuine long-range information flow.

**Questions:**

1. How do you ensure that high-entropy tokens truly correspond to informational gaps rather than discourse transitions or stylistic boundaries? Have you examined any qualitative examples of entropy peaks to confirm this assumption?


2. The paper treats entropy reduction as evidence of successful dependency capture. Could you clarify whether the reduction tends to occur at the lexical, syntactic, or semantic level? Have you compared this with other dependency metrics (e.g., attention shifts or mutual information)?

3. Does the entropy-based selection bias the resulting corpus toward more homogeneous or easier-to-predict contexts? Have you evaluated the lexical or topical diversity of the constructed dataset versus the raw corpus?

4. Given that each sample requires per-candidate forward verification, how does the total cost compare to standard long-context pretraining? Are there any approximations (e.g., smaller proxy models, cached logits) that could make the process scalable?

5. The experiments focus on a single model (Llama-3-8B). Do you expect the same entropy distribution patterns and improvements to hold for larger or smaller architectures, or for instruction-tuned models?

---

> ### Author Response · Authors · 2025-11-20
>
> **Questions 1.** How do you ensure that high-entropy tokens truly correspond to informational gaps rather than discourse transitions or stylistic boundaries? Have you examined any qualitative examples of entropy peaks to confirm this assumption?
>
> **Answer 1:**
>
> We consider that high-entropy tokens whose entropy can be reduced through retrieved context signal an information deficit. To address the reviewer’s question, we use Gemini 2.5 Flash to analyze the relationship between verified context document and the corresponding high-entropy tokens. Among tokens whose entropy could be reduced through retrieved context, more than 70% showed clear semantic or lexical dependencies on the added information.
>
> We categorize the relationship between verified context document and their corresponding high-entropy tokens into two broad classes: related and unrelated. The related class is further divided into six categories: Semantic Elaboration, Topical Consistency, Contextual Background, Factual Support, Narrative Continuation, and Syntactic Linking. The unrelated class consists of two categories: Unrelated and Contradictory. We sampled 1,036 verified context document–high-entropy token pairs (selecting the document that yielded the largest entropy reduction), and the detailed distribution is shown in the table below.
>
> | Category          | Count | Percentage (%) |
> |----------------------------|-------|----------------|
> | **Positive (73.55%)** |       |                |
> | Semantic Elaboration       | 268   | 25.87          |
> | Topical Consistency        | 256   | 24.71          |
> | Contextual Background      | 146   | 14.09          |
> | Factual Support            | 84    | 8.11           |
> | Narrative Continuation     | 6     | 0.58           |
> | Syntactic Linking          | 2     | 0.19           |
> | **Negative (26.45%)** |       |                |
> | Unrelated                  | 265   | 25.58          |
> | Contradictory              | 9     | 0.87           |
> | **Total Items**            | 1036  | 100.00         |
>
> **Questions 2-1.** The paper treats entropy reduction as evidence of successful dependency capture. Could you clarify whether the reduction tends to occur at the lexical, syntactic, or semantic level?
>
> **Answer 2-1:**
>
> For verified high-entropy tokens, we use Gemini 2.5 Flash to analyze whether the observed entropy reduction occurs at the lexical, syntactic, or semantic level. The breakdown of verified high-entropy tokens is as follows:
>
> Semantic-Level Reduction: 717 (71.70%)
>
> Lexical-Level Reduction: 257 (25.70%)
>
> Syntactic-Level Reduction: 26 (2.60%)
>
> These results indicate that most verified entropy peaks correspond to genuine missing information, with semantic dependencies being the dominant source.
>
> **Questions 2-2.** Have you compared this with other dependency metrics (e.g., attention shifts or mutual information)?
>
> **Answer 2-2:**
>
> We have not directly compared entropy-based dependency with other metrics such as attention shifts or mutual information. However, we analyzed the attention patterns of long-context models trained on our dataset. The results and detailed discussion can be found in Section 6.6.
>
> **Questions 3-1.** Does the entropy-based selection bias the resulting corpus toward more homogeneous or easier-to-predict contexts?
>
> **Answer 3-1:**
>
> No, the entropy-based selection does not bias the resulting corpus toward more homogeneous or easier-to-predict contexts. Our construction pipeline ensures this for two reasons:
>
> Stage 1 — Entropy reduction only verifies a possibility, not a guarantee of ease. Verified contexts simply indicate that if the model can retrieve the correct information from the global context, entropy at that position can be reduced. However, this reduction only happens after the model successfully establishes the long-range dependency. Without that ability, the sequence remains difficult.
>
> Stage 2 — Final training samples integrate many supplementary segments from different high-entropy positions. These segments are selected from different entropy peaks, not from thematically similar or topically homogeneous regions. This means the global sequence contains many distinct dependency relationships scattered across the context, and the model must learn to retrieve the corresponding pieces to benefit from them.
>
> Verified contexts indicate potential entropy reduction, but the model still has to effectively utilize the relevant long-range information for that reduction to occur; if it cannot perform long-context modeling, the sequence remains hard. Thus, the corpus is not biased toward surface-level easy or homogeneous text.

---

> ### Author Response · Authors · 2025-11-21
>
> **Questions 3-2.** Have you evaluated the lexical or topical diversity of the constructed dataset versus the raw corpus?
>
> **Answer 3-2:**
>
> For topical diversity, since the root documents are randomly sampled from the original pretraining corpus, their topic distribution naturally aligns with that of the source data.
>
> For lexical diversity, we computed the MaasTTR[1] metric and obtained comparable values: 0.0381 for the pretraining corpus and 0.0391 for our synthesized long-context dataset, indicating that lexical variety is well preserved.
>
> Additionally, the model maintains strong short-context performance, suggesting that the constructed dataset does not introduce substantial lexical or topical deviations from the original corpus.
>
> **Questions 4-1.** Given that each sample requires per-candidate forward verification, how does the total cost compare to standard long-context pretraining?
>
> **Answer 4-1:**
>
> We decompose the cost of long-context pretraining into two parts: (1) data construction and (2) model training, and compare our method against nextlong, the second-best baseline reported in the paper. We fully reproduced the data construction pipeline of nextlong for a fair comparison.
>
> In the data construction stage, EntropyLong incurs approximately 355 GPU hours relative to nextlong, primarily due to per-candidate verification. However, to reach the same RULER performance level as nextlong, EntropyLong requires substantially fewer training tokens, reducing the training cost by 307 GPU hours. Overall, the total computational cost of EntropyLong is therefore comparable to nextlong.
>
> | Method      | Embedding Computation | High-Entropy Identification | Verification | Training Cost (to RULER ≈85) | Total GPU Hours |
> |-------------|------------------------|-----------------------------|--------------|------------------------------|------------------|
> | nextlong    | 928 GPUh               | –                           | –            | 768 GPUh (4B tokens)         | 1696 GPUh    |
> | EntropyLong | 928 GPUh               | 3 GPUh                      | 352 GPUh     | 461 GPUh (2.4B tokens)       | 1744 GPUh    |
>
>
> **Questions 4-2.**  Are there any approximations (e.g., smaller proxy models, cached logits) that could make the process scalable?
>
> **Answer 4-2:**
>
> In contrast to full-scale pretraining, long-context extension typically does not rely on massive datasets. For instance, NVIDIA[2] uses fewer than 20B tokens and Kimi-K2[3] uses about 60B tokens for long-context training, indicating that the amount of long-context data construction required to support a SOTA model is not very high.
>
> Since our method is inherently model-in-the-loop, aiming to strengthen long-range dependencies for the specific model being extended, the most reliable practice is to compute entropy and verification using the target model itself. Although smaller proxy models or cached logits may serve as potential approximations, their utility remains speculative and they may not capture the dependency patterns most relevant for the model undergoing long-context extension. These approximation strategies may be explored in future work.
>
> **Questions 5.** The experiments focus on a single model (Llama-3-8B). Do you expect the same entropy distribution patterns and improvements to hold for larger or smaller architectures, or for instruction-tuned models?
>
> **Answer 5:**
>
> We expect that the general entropy distribution patterns identified in our method will be beneficial across different model architectures. However, we also acknowledge that each model may have its own optimal entropy thresholds due to differences in scaling and the heterogeneity of pre-training corpora.
>
> To examine cross-model applicability, we applied the same entropy-based data construction settings to two additional models, Llama-3-8B-Instruct and Llama-3.2-3B, and compared their performance with NExtLong. The results in the tables below show that EntropyLong consistently outperforms NExtLong across all context lengths, demonstrating strong transferability of the proposed approach.
>
> Llama-3-8B-Instruct
> | method      | 8k    | 16k   | 32k   | 64k   | 128k  | avg   |
> | ----------- | ----- | ----- | ----- | ----- | ----- | ----- |
> | entropylong | 93.50 | 92.43 | 88.91 | 85.07 | 82.95 | 88.57 |
> | nextlong    | 92.39 | 90.41 | 87.72 | 82.69 | 80.96 | 86.84 |
>
>
> Llama-3.2-3B
> | method      | 8k    | 16k   | 32k   | 64k   | 128k  | avg   |
> | ----------- | ----- | ----- | ----- | ----- | ----- | ----- |
> | entropylong | 80.43 | 78.83 | 73.71 | 70.78 | 64.12 | 73.57 |
> | nextlong    | 79.33 | 77.98 | 71.69 | 70.34 | 62.35 | 72.34 |
>
>
>
> [1] McCarthy P M, Jarvis S. MTLD, vocd-D, and HD-D: A validation study of sophisticated approaches to lexical diversity assessment[J].
>
> [2] Nano N N. Efficient hybrid mamba-transformer reasoning model[J]. arXiv preprint arXiv:2508.14444, 2025.
>
> [3] Team K, Bai Y, Bao Y, et al. Kimi k2: Open agentic intelligence[J]. arXiv preprint arXiv:2507.20534, 2025.

---

> > ### Author Response · Authors · 2025-11-23
> >
> > **Supplementary Analysis for Questions 4-1**
> >
> > Previous analysis demonstrated that EntropyLong achieves better token efficiency than NExtLong, for a further fair comparison, we investigate the scenario where NExtLong’s total training tokens are increased to 6B.
> >
> > For reference, training EntropyLong on its original 4B-token dataset costs a total of 2051 GPU hours (embedding computation 928 GPUh + high-entropy identification 3 GPUh + verification 352 GPUh + training 768 GPUh). Under the expanded 6B-token setting, NExtLong’s total GPU cost reaches 2080 GPU hours, slightly more than EntropyLong .
> >
> > Under this setting, NExtLong’s RULER performance reaches 85.75, which remains lower than EntropyLong’s 87.37 achieved with the original 4B-token training. As shown in Table below, EntropyLong achieves higher performance at longer context lengths (32k to 128k).
> >
> > The results show that, when aiming for better performance—which typically needs more computational resources—EntropyLong demonstrates higher computational efficiency compared to previous methods.
> >
> > | Method                     | 8k     | 16k    | 32k     | 64k     | 128k    | Avg    | Total GPU Hours |
> > |----------------------------|--------|--------|---------|---------|---------|--------|----------------|
> > | NExtLong (6B tokens)     | **91.97** | **90.52** | 85.81   | 81.88   | 78.60   | 85.75  | 2080           |
> > | EntropyLong (4B tokens)    | 91.50  | 90.11  | **88.95** | **85.04** | **81.26** | **87.37** | 2051           |

---

> ### Author Response · Authors · 2025-11-21
>
> **Weaknesses 1.** The central assumption is that high predictive entropy indicates missing long-range information, which lacks strong theoretical grounding. Empirically, many high-entropy tokens correspond to discourse openings, transitions, or rare words rather than genuine dependency points. Thus, the “verified” concatenations may enhance topical smoothness rather than causal linkage.
>
> **Answer W1:**
>
> We note that in our method, only the high-entropy positions that pass the verification step are used as potential long-range dependency signals, as described in Section 3.2 and further validated in Section 6.1. We do not assume that all high-entropy tokens reflect missing information. We acknowledge that many high-entropy positions arise from discourse openings, transitions, or rare words. However, the verification step helps separate genuine information gaps from natural ambiguity or structural variation.
>
> In our response to Question 1, we discussed how the verified high-entropy positions relate to their retrieved supplementary documents. We also examined high-entropy positions that failed verification, meaning their entropy did not decrease after adding context. Using Gemini 2.5 Flash, we categorized these cases.
>
> The results are as follows:
> - Inherent Diversity (Open Choice): 1022 (51.10%)
> - Transitional or Structural Boundary: 723 (36.15%)
> - Information Gap: 255 (12.75%)
>
> These findings show that many high-entropy tokens that do not pass the verification step arise from inherent diversity, discourse transitions, or structural boundaries rather than from genuine missing information. This supports the effectiveness of the verification step. The verification step filters out these cases and keeps positions that are meaningfully improved by additional context.
>
> **Weaknesses 2.** The paper does not analyze what types of dependencies are actually captured after verification (e.g., factual consistency, narrative continuity, or syntactic linking). Without qualitative inspection or causal tracing, the claim of “true long-range reasoning” remains speculative.
>
> **Answer W2:**
>
> We discussed the types of dependencies captured after verification in our response to Question 1.
>
> **Weaknesses 3.** The framework requires per-sample forward passes for entropy computation and re-verification, which substantially increases computational cost. The paper does not discuss efficiency trade-offs or scaling to larger corpora.
>
> **Answer W3:**
>
> We discussed the cost in our response to Question 4-1.
>
> **Weaknesses 4.** Although the paper frames its contribution as constructing long-context dependencies, the retrieval query for each high-entropy token is limited to a 16-token local window. This narrow context restricts semantic scope to near-sentence continuations rather than true cross-document or distant dependencies, effectively reducing the retrieval process to localized topical or lexical matching. As a result, the constructed samples may improve surface fluency but do not necessarily capture genuine long-range information flow.
>
> **Answer W4:**
>
> As discussed in our response to Question 1, the 16-token local window is already sufficient to retrieve context segments that exhibit clear dependency relations. In addition, both our long-context and short-context evaluation results support the effectiveness of this design choice.

---

> ### Author Response · Authors · 2025-11-27
>
> Dear Reviewer dtra,
>
> We sincerely appreciate the insightful feedback you provided, which has been instrumental in enhancing the quality of our work. We have prepared detailed responses for your review.
>
> Please let us know if we need to provide any further clarifications or feedback.
>
> Best regards,
>
> The Authors

---

### Author Response · Authors · 2025-11-30
**General Response by Authors**

**Dear Reviewers, ACs, and SACs,**

We sincerely thank all reviewers for the thorough and insightful feedback. These constructive suggestions and detailed questions have directly helped us improve and clarify the paper. We are grateful that the reviewers recognize EntropyLong as **a novel and effective approach for synthesizing long-context data**. We are encouraged by the following positive remarks from the reviews:

1. **Innovation of method and idea.** Reviewers highlighted that the paper proposes a new, model-informed approach to constructing long-context data that goes beyond heuristic concatenation and better supports long-range dependency learning. (**Reviewer dtra, PDbx and L2EC**)

2. **Clear motivation and validated assumptions.** Reviewers agreed that model predictive uncertainty is a meaningful signal for identifying informative long-distance dependencies, and appreciated the extensive experiments validating this hypothesis. (**Reviewer 2y86, PDbx and L2EC**)

3. **Strong and convincing experiments.** Reviewers acknowledged the consistent improvements over baselines (**+2.15 on RULER and +3.5 on LongBenchv2** over the second-best method) and commended the rigor of our ablation studies and analyses (**Reviewer dtra and L2EC**).

4. **Research insight and potential impact.** Reviewers noted that our work opens new directions in uncertainty-guided data modification and provides insights into enhancing long-context understanding in language models. (**Reviewer PDbx**)

We deeply appreciate the constructive suggestions, which significantly strengthened the paper. In response to reviewers’ feedback, we have made the following improvements:

1. Added an analysis of the cost of data construction, demonstrating higher token efficiency. (**Appendix E**)

2. Added an analysis of the relationship between high-entropy positions and the supplemental context, and examined how the entropy distribution changes after adding supplemental text, providing insight into the categories of high-entropy tokens. (**Appendix K, L**)

3. Discussed how dependency distance affects long-range performance and validated this with detailed experiments. (**Appendix J**)

4. Analyzed the generality of EntropyLong across architectures, model sizes, and instruct models, supported by experiments. (**see discussion with L2EC**)

5. Added comparisons with additional baselines (UTK and Llama3.1-128k), further demonstrating the effectiveness of EntropyLong. (**Appendix G and discussion with 2y86**)

6. Added discussion on the trade-off between the α and σ thresholds, strengthening the contribution. (**Appendix I**)

7. Improved clarity in describing the details of the data construction process. (**Appendix B**)

After actively engaging in discussions with the reviewers and incorporating the improvements listed above, reviewer PDbx (original rating: 4) indicated a positive inclination toward **raising the score**, and reviewer L2EC (original rating: 6) noted that **“most of my questions and weaknesses have been addressed.”**.

We hope this summary of the reviewers’ positive feedback, their suggestions, and our corresponding revisions is helpful for the ACs. We sincerely thank all Reviewers, ACs, and SACs for their hard work and thoughtful comments, which have made this paper more rigorous and complete. We are grateful for the encouragement and guidance provided throughout the review process.

**Sincerely,**

**Authors**

---

### Meta-Review · Area_Chair_xeK3 · 2025-12-29

**Summary:**

Overall, the reviewers agreed that EntropyLong presents a creative and well-executed approach to constructing long-context data using model-predicted entropy as a guiding signal, and that it delivers clear empirical benefits on strong benchmarks such as RULER and LongBench v2. The paper was recognized for its novelty, methodological rigor, and detailed experiments.

However, there remained some divergence in the depth of theoretical and conceptual validation. While reviewers 2y86, L2EC, and PDbx acknowledged the added analyses, efficiency discussion, and dependency categorization, reviewer dtra continued to question whether high entropies truly represent missing long-range information rather than linguistic variability.

After checking the quality of the paper, we believe the consensus leaned toward a borderline but positive recommendation. This work’s empirical results and methodological contributions are significant, even though its theoretical underpinnings and large‑scale scalability require further substantiation.

**Reviewer Concerns:**

Concerns Effectively Addressed by the Rebuttal
- The rebuttal clarified the empirical validity of high‑entropy positions, showing that more than 70% of verified cases correspond to genuine semantic or factual dependencies, effectively addressing doubts about whether entropy reflects missing information.
- Qualitative and quantitative analyses were added, illustrating the types of dependencies captured and providing examples of entropy reduction patterns.
- The authors provided a clear computational cost comparison against NExtLong, demonstrating that despite verification overhead, the approach achieves higher token efficiency and comparable total GPU hours.
- The rebuttal added more empirical results by experimenting with additional baselines, backbone models, and extended ablation studies exploring the effects of entropy thresholds, document placement, and ordering.

Concerns Still Outstanding or Partially Addressed
- The paper still lacks a formal theoretical justification connecting token‑level entropy to genuine causal or discourse‑level long‑range dependencies; the argument remains empirical.

- Scalability to much larger corpora or trillion‑token regimes remains uncertain. In addition, the feasibility of applying the pipeline at full pre‑training scale was not demonstrated.

**Reviewer Scores:**

Reviewer 2y86 and Reviewer L2EC were satisfied after the rebuttal, noting that most of their concerns had been thoroughly addressed.

Reviewer PDbx, who initially gave a marginal score (4), indicated an intention to raise it to (6) after finding the authors’ clarifications and additional analyses convincing.

Reviewer dtra maintained a reject score (2) due to continuing doubts about the theoretical soundness of the approach and the authenticity of the claimed long‑range dependencies.

---

### Decision · Program_Chairs · 2026-01-26

Accept (Poster)